# Identification of distinct cytotoxic granules as the origin of supramolecular attack particles in T lymphocytes

Hsin-Fang Chang [1,10 ✉], Claudia Schirra[1,10], Momchil Ninov [2,3,4], Ulrike Hahn[1], Keerthana Ravichandran[1], Elmar Krause [1], Ute Becherer [1], Štefan Bálint [5], Maria Harkiolaki [6], Henning Urlaub [2,4], Salvatore Valitutti [7,8], Cosima T. Baldari[9], Michael L. Dustin [5], Reinhard Jahn [3] & Jens Rettig[1 ✉]

Cytotoxic T lymphocytes (CTL) kill malignant and infected cells through the directed release of cytotoxic proteins into the immunological synapse (IS). The cytotoxic protein granzyme B (GzmB) is released in its soluble form or in supramolecular attack particles (SMAP). We utilize synaptobrevin2-mRFP knock-in mice to isolate fusogenic cytotoxic granules in an unbiased manner and visualize them alone or in degranulating CTLs. We identified two classes of fusion-competent granules, single core granules (SCG) and multi core granules (MCG), with different diameter, morphology and protein composition. Functional analyses demonstrate that both classes of granules fuse with the plasma membrane at the IS. SCG fusion releases soluble GzmB. MCGs can be labelled with the SMAP marker thrombospondin-1 and their fusion releases intact SMAPs. We propose that CTLs use SCG fusion to fill the synaptic cleft with active cytotoxic proteins instantly and parallel MCG fusion to deliver latent SMAPs for delayed killing of refractory targets.

[1] Cellular Neurophysiology, Center for Integrative Physiology and Molecular Medicine (CIPMM), Saarland University, 66421 Homburg, Germany. [2] Bioanalytical Mass Spectrometry, Max Planck Institute for Multidisciplinary Sciences, Am Fassberg 11, 37077 Göttingen, Germany. [3] Laboratory of Neurobiology, Max Planck Institute for Multidisciplinary Sciences, Am Fassberg 11, 37077 Göttingen, Germany. [4] Bioanalytics, Institute for Clinical Chemistry, University Medical Center Göttingen, Robert Koch Str. 40, 37075 Göttingen, Germany. [5] Kennedy Institute of Rheumatology, Nuffield Department of Orthopaedics, Rheumatology and Musculoskeletal Sciences, University of Oxford, OX3 7FY Oxford, UK. [6] Diamond Light Source, Harwell Science and Innovation Campus, OX11 0DE Didcot, UK. [7] Cancer Research Center of Toulouse, INSERM U1037, 31037 Toulouse, France. [8] Department of Pathology, Institut Universitaire du Cancer-Oncopole de Toulouse, Toulouse, France. [9] Department of Life Sciences, University of Siena, 53100 Siena, Italy. [10] These authors contributed equally: Hsin-Fang Chang, Claudia Schirra. ✉email: Hsin-Fang.Chang@uks.eu; jrettig@uks.eu

Cytotoxic T lymphocytes (CTL) fight against pathogens and cancer by eradicating infected or transformed cells, referred to collectively as target cells. Target cell killing occurs at a specialized contact zone called the immunological synapse (IS), by the regulated exocytosis of cytotoxic granules (CG)[1–3]. CGs contain perforin-1 (Prf1) and granzyme B (GzmB) and the membrane anchored Fas-Ligand (FasL) that induce target cell death by apoptosis and necrosis[4,5]. The fusion of CGs with the plasma membrane at the IS is mediated by a specific SNARE complex consisting of the t-SNAREs Syntaxin11 and SNAP-23 on the plasma membrane and a v-SNARE on the CG membrane[6,7]. The v-SNARE of humane CTLs is VAMP7 and that of mouse CTLs is VAMP2 (aka synaptobrevin2, Syb2)[8,9].

CGs are lysosome-related cell-type specific organelles, which share features with endosomes and lysosomes[10,11]. The proteoglycan serglycin (Srgn) is required to form dense cores, named for their appearance in transmission electron microscopy, and enables retention of GzmB in CGs through its binding to dozens of GzmB molecules[12,13]. Serglycin-deficient CTLs display defective CG maturation, but only show mild defects in their killing capacity[14–16], indicating the existence of alternative killing pathways. Numerous morphological studies point to a hetero-geneity of Prf1- and GzmB-containing organelles[11,17–21]. This heterogeneity may indicate intermediate forms during biogenesis or it may indicate different functional CG types.

CTLs release Prf1 and GzmB in two forms[22]. Prf1 and GzmB disperse rapidly as soluble proteins that are immediately available to interact with the target cells, but are restricted to the IS to provide specificity[23]. Recent evidence indicates that about half of the released Prf1 and GzmB are found in supramolecular attack particles (SMAPs), which have a core of cytotoxic proteins and a shell of thrombospondin-1 (TSP-1)[22]. SMAPs remain active for hours and thus may act in a different time frame and outside of the IS[24]. The source of SMAPs within cytotoxic cells is unknown.

Here, we employ a synaptobrevin2-mRFP knock-in mouse[8] in an unbiased approach to purify mature, fusion-competent CGs from mouse CTLs. Using density gradient centrifugation followed by immuno-isolation and mass spectrometry we find two classes of fusion-competent CGs. Transmission and scanning electron microscopy analysis discerned different morphologies, with one CG class containing a single dense core and uniform diameter (single core granule, SCG) and the other class containing multiple cores and varying diameters (multi core granule, MCG). Mass spectrometry of purified granules shows that both granules contain large amounts of GzmB. However, they differ in that SCGs are lysosomal-like, containing cathepsins, while MCGs are more heterogeneous and contain endosomal-like proteins such as Rabs. Analyses by super-resolution and total internal reflection fluorescence microscopy demonstrates that both classes of CGs fuse at the IS and that MCGs are the source of SMAPs. Our results show an additional type of cytotoxic granule nearly 40 years after the discovery of SCG.

## Results

### CTLs from synaptobrevin2-mRFP knock-in mice as source for fusion-competent cytotoxic granules.

We previously generated a knock-in mouse in which a monomeric red fluorescent protein is fused to the C-terminus of the v-SNARE Syb2 (Syb2 KI)[8]. The CGs from Syb2 KI CTLs are labeled with mRFP, polarize to the IS upon target cell contact (Fig. 1a) and co-localize with GzmB (Fig. 1b). Because Syb2 is required to mediate the fusion of CG at the IS[8], CTLs derived from the Syb2 KI mouse provide a unique source for the selective purification of fusion-competent CGs.

We isolated CD8[+] T cells from Syb2 KI mice, activated them for 3 days and assessed their phenotype by flow cytometry. We

obtained 85% viable effector cells, of which all were Syb2-mRFP[+] (Supplementary Fig. 1a, b), 97% were CD44[+], 41.0% were CD62L[−] effectors and 89% of CD25[+] activated cells (Supplementary Fig. 1c–e). After verifying the high yield of CTLs, we used nitrogen cavitation to crack the cells open, removed the cell debris by low-speed centrifugation, and fractionated the supernatant by ultracentrifugation on a discontinuous sucrose density gradient (Fig. 1c). Fluorescently labeled particles were concentrated at the interfaces between 1.0 and 1.2 M, and 1.2 and 1.4 M sucrose named fractions 6 and 8, respectively (Fig. 1c bottom). These fractions were enriched with the CG protein GzmB (Fig. 1d) and contained intact CGs (Fig. 1e).

For further purification, we incubated the granule-enriched gradient fractions 6 and 8 with magnetic beads containing an immobilized monoclonal antibody specific for Syb2 (Fig. 2a). Western blot analysis of the immuno-isolated material showed an enrichment of not only Syb2 but also the CG content proteins GzmB (Fig. 2b) and Prf1 (Supplementary Fig. 2a). On the other hand, contaminating membranes such as plasma membrane fragments, identified by Na[+]/K[+]-ATPase, remained in both supernatants (Fig. 2b). In order to confirm that the purified granules were still intact after immuno-isolation we crossed bred the Syb2 KI mice with our recently reported GzmB-mTFP knock-in (GzmB KI) mice[25] and repeated the entire procedure of ultracentrifugation and immuno-isolation with CTLs derived from these double KI (Syb2/GzmB DKI) mice. Confocal microscopy showed a strong co-localization of both fluorophores (Fig. 2c). Object based co-localization algorithm determined that $66.1 \pm 0.6\%$ Syb2 mRFP[+] spots co-localized with GzmB-TFP and that $83.3 \pm 0.5\%$ TFP[+] spots co-localized with mRFP ($n = 3$, mean ± SD). This demonstrated that the GzmB was still contained within the granules (Fig. 2c). Correlative light and electron microscopy (CLEM) of the immuno-precipitates of fractions 6 (IP6) and 8 (IP8) further confirmed the co-localization of Syb2 and GzmB and the integrity of the granule membrane (Fig. 2d).

### CTLs contain two classes of exocytosis-competent granules with different morphology.

We next visualized the morphology of the granules contained in IP6 and IP8 by electron microscopy. Scanning electron microscopy in secondary electron mode (Fig. 3a, left) and back-scattered electron mode (Fig. 3a, right) showed that granules from both IPs had a round, smooth appearance on the Dynabeads, with granules from IP6 being slightly larger in diameter than granules from IP8. Transmission electron microscopy on these IP samples fixed by high-pressure freezing showed that IP8 contained a homogeneous population of granules with a single dense core occupying the entire lumen of the granule (Fig. 3b, lower row), reminiscent of the classical CGs described in the literature[11,20]. In contrast, granules from IP6 were quite variable in their size and composition, containing two or more particles with dense core-like structures and several other less electron dense particles (Fig. 3b, upper row; Supplementary Fig. 3a). Based on their appearance in electron microscopy we refer to granules from IP6 as multi core granules (MCGs) and to granules from IP8 as single core granules (SCGs) (Fig. 3c). Quantitative analysis of granule diameter yielded an average diameter of $364 \pm 12$ nm (mean ± sem; $N_{IP} = 3$, $n_{granules} = 118$) for MCGs and $293 \pm 8$ nm for SCGs (mean ± sem; $N_{IP} = 3$, $n_{granules} = 85$; Fig. 3c). As shown in Fig. 3b the morphological appearance varied particularly for MCGs from IP6. Therefore, we further analyzed the different observed variants. While more than 80% of granules from IP8 contained a single dense core, almost 60% of granules from IP6 contained multiple dense cores and more than 20% contained several vesicles surrounded by bilayer membranes (Fig. 3d). We also noticed that several particles

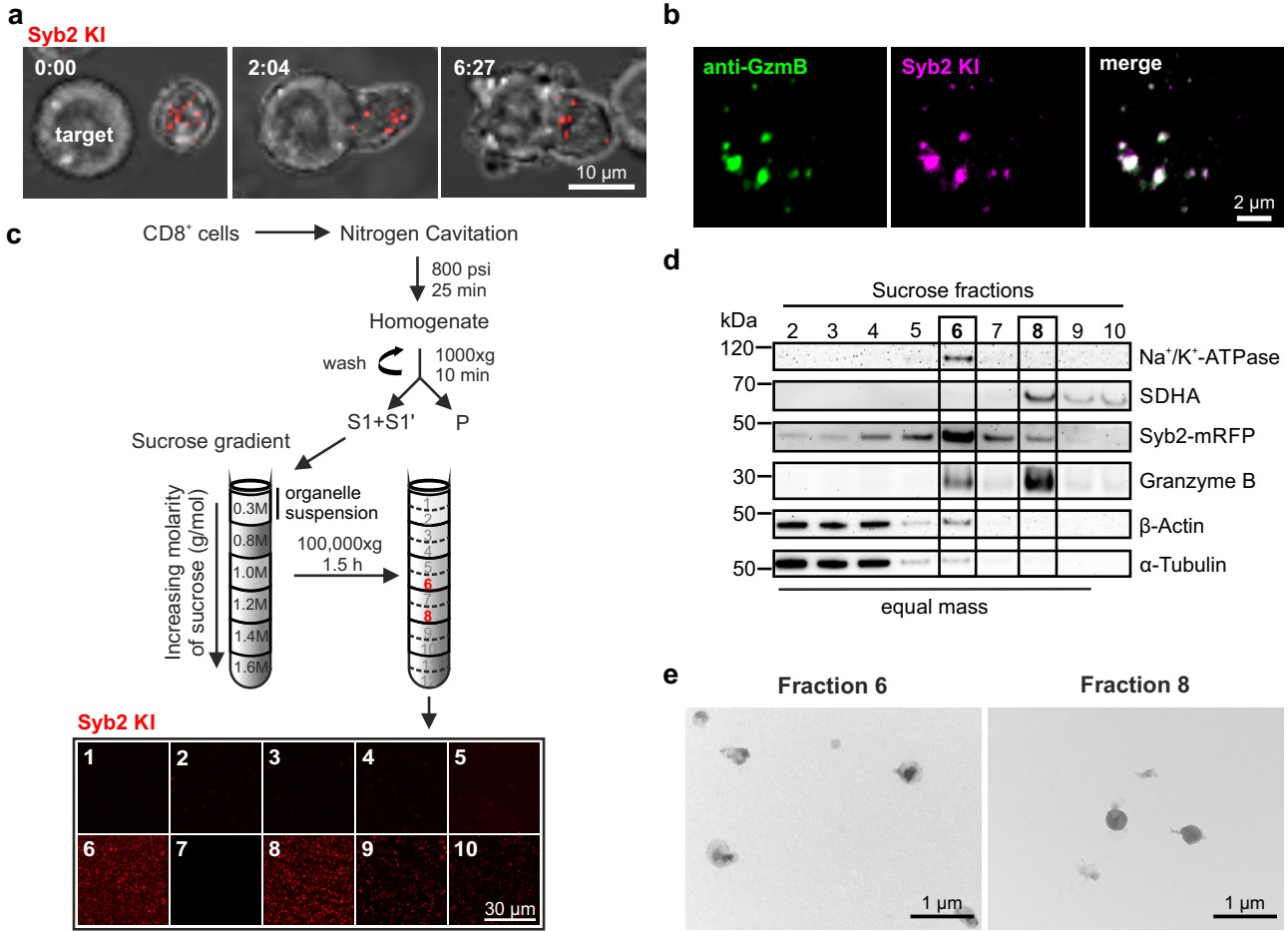

**Fig. 1 Cytotoxic granules (CGs) of CTLs from synaptobrevin2-mRFP knock-in (Syb2 KI) mice can be visualized after subcellular fractionation.**
**a** Confocal time-lapse images of a representative activated cytotoxic T lymphocytes (CTLs) derived from Syb2 KI mice in contact with a target cell. Endogenously labeled CGs (red, left) polarized to the immunological synapse (IS) upon T cell-target cell contact (middle) and lead to target cell death (right). $N_{mouse} = 6$; $n_{cells} = 31$. **b** Representative structured illumination microscopy (SIM) images of a CTL isolated from Syb2 KI mouse with mRFP (magenta) labeled CGs. Cells were stained with an anti-granzyme B (GzmB) antibody coupled to Alexa647 (green). $N_{mouse} = 1$; $n_{cells} = 33$. **c** Flow chart for CG isolation procedure based on discontinuous sucrose density gradient centrifugation. Ten 1 ml fractions (1–10) were collected from the gradient and pelleted on glass coverslips. Confocal images of the fractions are shown below. $N_{gradient} = 1$; $n_{images} = 2$. **d** Quantitative western blot for sucrose fractions 2–10 (0.7 μg protein per lane, fraction 10 with max. volume of 30 μl) probed against proteins that reside on CGs (Syb2-mRFP and GzmB), plasma membrane (Na$^+$/K$^+$-ATPase), mitochondria (SDHA; technical replicate) and cytoskeleton (β-actin (reprobed) and α-tubulin). $N_{gradients} = 3$.
**e** Transmission electron microscopy (TEM) images of sucrose gradient fractions 6 and 8. $N_{gradients} = 3$; $n_{images} = 20$ per fraction. Source data are provided as a Source data file.

contained in MCGs were not surrounded by a lipid membrane (Supplementary Fig. 3), reminiscent of the recently reported SMAPs[22], and we refer to these non-membrane bound electron dense particles as SMAP-like for now. Detailed analysis of SMAP-like containing MCGs gave an average particle diameter of 122 ± 7 nm (mean ± sem; $n = 74$; Fig. 3e), in excellent agreement with the published value of 120 ± 43 nm for SMAPs in human primary CTLs[22]. Additionally, using Cryo-Soft X-Ray tomography, we observe that both types of organelles with either the classical dense core or the multicores are also contained in human natural killer cell line NK92[26] (Supplementary Fig. 4, Supplementary Movies 1 and 2). Hence, we conclude that cytotoxic cells contain two different classes of fusion-competent CGs- SCGs and MCGs, and that SMAPs might be stored in MCG before release.

**Multi core and single core granules have different protein compositions.** To learn about the composition of MCGs and

SCGs, we subjected IP6 and IP8 to mass spectrometry analysis. Because naive CD8$^+$ T cells do not contain CGs, we used immuno-precipitates of naïve CD8$^+$ T cells as negative control (Supplementary Fig. 2b, c). Label-free quantification (LFQ) of five and three biological and technical replicates of IP6 and IP8, respectively, showed a very high reproducibility (Supplementary Fig. 5a, b). For MCG we identified a total of 212 proteins, while for SCG a total of 384 proteins could be detected. Interestingly, only 156 proteins were found to be present in MCG and SCG (Fig. 4a, Supplementary Data 1). Both granule types contained GzmB and Syb2-mRFP confirming their cytotoxic and exocytotic competence. We also found LAMP-1 and almost all the subunits of vesicular H$^+$ATPase (Fig. 4b, Supplementary Fig. 5c, d). The latter is required to generate the granule's intra-vesicular acidic environment[27]. This was in a good agreement with similar labeling of MCGs and SCGs with LysoTracker (Supplementary Fig. 6). Prf1 was not significantly enriched nevertheless, three peptides of Prf1 were detected in 2 out of 6 measurements in IP8 (7% coverage). The presence of Prf1 was further verified by

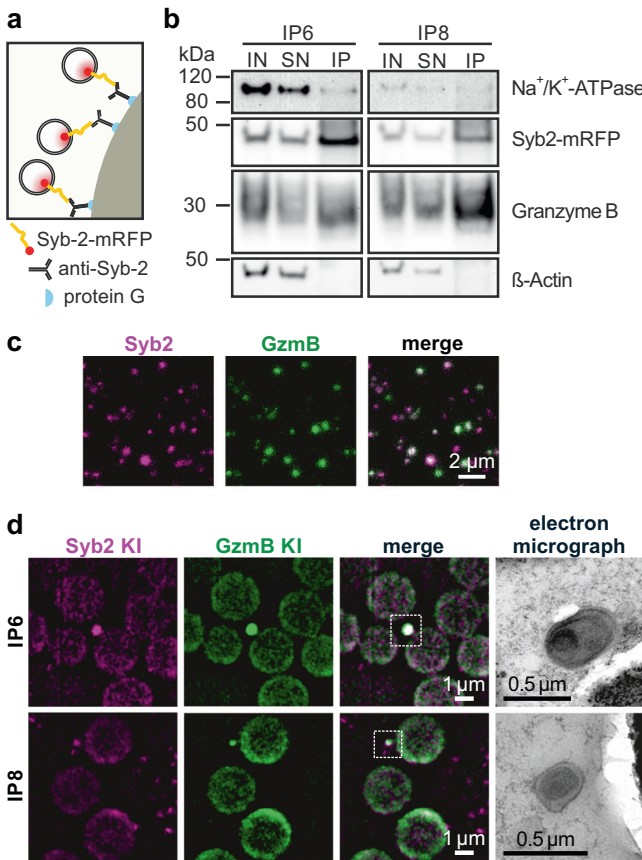

**Fig. 2 Immuno-isolation of two classes of cytotoxic granules from CTLs using an anti-synaptobrevin2 (Syb2) antibody. a** Scheme of Syb2-mRFP positive cytotoxic granules bound to the antibody-conjugated Dynabeads Protein G following immunoprecipitation (IP). **b** Representative western blots of IP6 and IP8 with input (IN, 2%), supernatant (SN, 2%) and IP (40%) volume. Protein markers for plasma membrane (Na$^+$/K$^+$-ATPase), cytotoxic granules (Syb2-mRFP, GzmB) and cytoskeleton (β-actin) were used (N$_{IP}$ = 4). **c** 500 μl of sucrose fraction 6 and 8 from Syb2/GzmB DKI CTLs were pooled and centrifuged on glass coverslip (N$_{gradient}$ = 1, n$_{images}$ = 3). The confocal images are shown as single channel (green and magenta) and merge image. **d** Correlative light and electron microscopy (CLEM) images of immuno-isolated organelles of Syb2/GzmB DKI CTLs (N$_{IP}$ = 1, n$_{images}$ = 7 and 4 for IP6 and IP8, respectively). Left to right: SIM fluorescence images of IPs are shown for Syb2 (magenta) and GzmB (green), and the corresponding organelle is represented in the TEM images. Upper and lower panel represent IP6 and IP8, respectively. Source data are provided as a Source data file.

immunoblotting in IP6 and IP8 (Supplementary Fig. 2a). The SCG fraction was lysosomal-like with cathepsins as a marker while the MCG fraction contained a number of recycling plasma membrane proteins and a specific subset of Rab GTPases associated with endosomal-like compartments (Fig. 4b, Supplementary Fig. 5c, d). Notably, MCG also contained proteins previously associated with SMAPs by mass spectrometry analysis (e.g., RPL12, HSP90b1) (Fig. 4b). While released SMAPs contained few membrane proteins[22], the MCGs naturally contain many membrane proteins associated with their limiting membrane and intraluminal vesicle membranes. Sunburst plots of gene ontology analysis for both fractions showed clear differences in putative subcellular localization and biological function (Fig. 4c, d). These data indicate that these two granule types represent distinct steady state compartment, rather than different stages of a single maturation pathway.

**Supramolecular attack particles are released from multi core granules.** After identifying putative SMAPs in MCGs, we examined whether we could visualize their release. For that purpose, we seeded CTLs derived from wild-type and GzmB KI mice for 90 min on supported lipid bilayer (SLB) containing anti-CD3 antibody, thus stimulating SMAP release[22]. CTLs were then fixed in place on the SLB and stained with anti-GzmB antibody. Structured illumination microscopy (SIM) indicated the presence of GzmB$^+$ particles on the SLB associated with and away from cells (Fig. 5a, arrows). This is consistent with SMAPs being released by CTLs that subsequently migrated away from the initial attachment site, leaving SMAPs behind. Because TSP-1 is a component of SMAP's shell we sought to provide additional evidence for SMAPs in MCGs by expressing TSP-1-GFPspark together with GzmB-mCherry fusion constructs in CTLs. SIM images of fixed, double-transfected CTLs clearly demonstrated a partial co-localization of TSP-1 and GzmB defining the MCG compartments, whereas granules that are only GzmB$^+$ are SCGs (Fig. 5b). Furthermore, we observed that TSP-1 was highly enriched in isolated MCG population rather than in SCG (Supplementary Fig. 7). This data strengthens our findings that MCGs and SCGs carry distinct cargos. Additionally, we took advantage of the fact that wheat germ agglutinin (WGA) specifically binds to glycoproteins present in SMAPs[22]. CTLs from GzmB KI were pre-incubated with WGA-Alexa647 for 90 min to allow uptake in SMAPs and seeded on poly-ornithine or anti-CD3 coated coverslips, for resting and activated CTLs, respectively. We found that about 60% of granules were MCGs co-labeled with WGA and GzmB, 20% were SCGs only containing GzmB, while the rest were small single WGA-positive organelles distributed throughout the cell cytoplasm and at the IS (Fig. 5d). The size of these organelles was 410 ± 7 nm, while the SCGs were 322 ± 7 nm in diameter (Fig. 5e, mean ± sem; n = 275 for MCG and n = 156 for SCG in 20 cells). Note that the size distribution of MCGs and SCGs corresponded closely to the size distribution measured on IP6 and IP8 by EM (Fig. 3c). To investigate the secreted SMAPs we seeded CTLs labeled with WGA on SLBs. After additional 90 min, cells were washed away and the remaining SMAPs were stained with anti-TSP-1 antibody. The representative SIM images showed co-localization of GzmB, TSP-1 and WGA, which formed a ring around the GzmB and TSP-1 fluorescent signal (Fig. 5c). To verify their morphology in intact cells, we performed CLEM experiments on CTLs which had been co-transfected with GzmB-mCherry and TSP-1-GFPspark and incubated with WGA. The organelle in the electron micrograph correlating to fluorescent puncta in all SIM channels had indeed a similar morphology as isolated MCGs shown in Fig. 3b (Fig. 5f, right row). We also found granules, which contained only GzmB and were smaller with a more uniform contents, resembling SCGs. Furthermore, we detected smaller organelles containing only TSP-1. These data further establish that SMAPs are stored in MCGs.

To evaluate the potential killing efficiency of both granules we measured the amount of GzmB content. For that we visualized intact granules with STED microscopy from fraction 6 and 8 that were settled on glass coverslips. This allowed us to clearly resolve 2.5 ± 0.2 (n = 52) individual SMAPs per MCGs (Fig. 6a). Interestingly, it also showed that GzmB is not evenly distributed in the dense core of SCGs. Overall the GzmB fluorescence was 40% weaker and more diffuse in SCGs than in MCGs (p < 0.001, n = 52 and 44 for MCG and SCG, respectively; Fig. 6b). The fraction of the granule volume occupied by GzmB particles was slightly but not significantly higher in SCGs as compared to MCGs. Therefore, the overall concentration of GzmB, which is directly related to the GzmB fluorescence intensity in the entire granule volume, is higher in MCGs than in SCGs by 20%

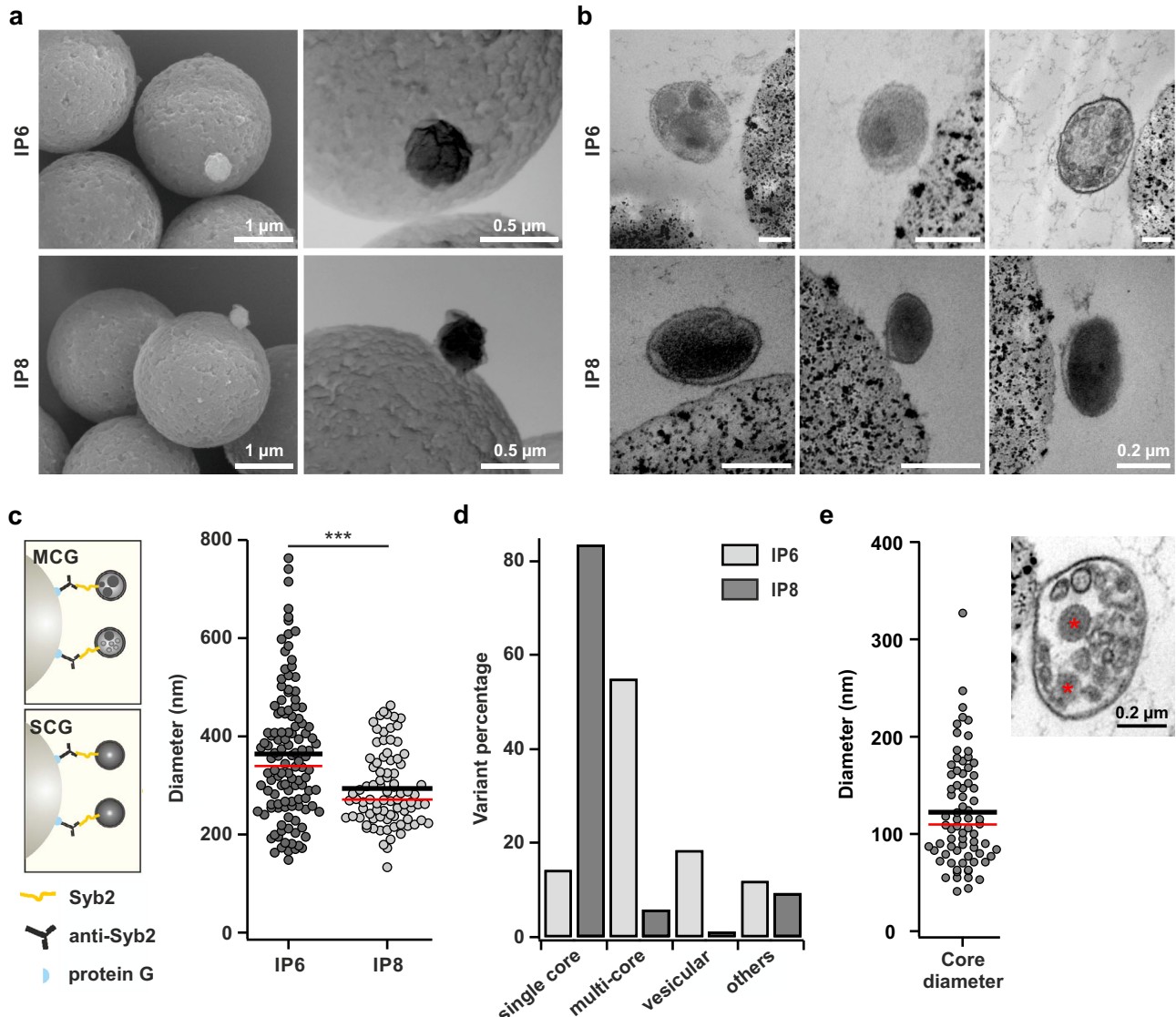

**Fig. 3 SEM/TEM analysis discerns two types of morphological distinct cytotoxic granules after immunoprecipitation. a** Representative scanning electron microscope (SEM) images of organelles derived from IP6 and IP8 following immunoprecipitation with anti-synaptobrevin2 antibody ($N_{IP} = 2$, $n_{granules} = 21$ and 19 for IP6 and IP8, respectively). Left: SEM image in secondary electron mode. Right: SEM image in back scattered mode. **b** Three representative TEM images of IP6 and IP8 display a heterogeneous population of multi core granules (MCG; upper row) and a homogenous population of single core granules (SCG; lower row), respectively. **c** Scheme of MCG and SCG (left) and diameter analysis (right) of MCGs in IP 6 ($N_{IP} = 3$, $n_{granules} = 118$) and SCGs in IP 8 ($N_{IP} = 3$, $n_{granules} = 85$, $p < 0.001$) as shown in (**b**). **d** Quantification of morphological variants of the organelles from IP6 and IP8 as shown in (**b**). Percentage of organelles classified into four groups based on their ultrastructural features: single core (uniformly electron dense), multicore (small dense cores with or without small vesicles), vesicular (many small vesicles) and others. The observed distribution of organelles in the four categories between IP6 and IP8 are significantly different ($\chi^2 = 102.73$ and df $= 3$, $p < 0.001$). **e** Quantification of dense core diameter observed in MCGs ($N_{IP} = 3$; $n_{granules} = 74$) as shown in (**b**). Inset: representative image of MCG containing small dense cores marked with asterisks. In all scatter dot plots black line represents mean and red line median. Source data are provided as a Source data file.

(Mann–Whitney $U$ test; $p = 0.003$, $n = 52$ and 44 for MCG and SCG, respectively; Fig. 6b). This analysis indicates that SMAPs are hot spots of cytotoxic proteins such as GzmB and that they are likely to be very effective in killing target cells.

To visualize fusion of SCG and MCG in real time we performed TIRF microscopy on CTLs seeded on SLBs. We again used CTLs derived from the GzmB KI mouse to endogenously label both granule classes and pre-incubated them for 90 min with WGA to label SMAPs. We frequently identified single CTLs releasing both SCGs and MCGs (Fig. 7a, c), with SCGs characterized by being GzmB-positive and WGA-negative (SCG fusion event at 39.2 s of Fig. 7a; Supplementary Movie 3) and

MCG by being double-positive for GzmB and WGA (fusion event at 78.4 s of Fig. 7a). The MCG fusion events displayed partial dissipation of TFP from the fusion sites, which might represent the release of a small amount of the cleaved TFP from GzmB[25] (Fig. 7b). Nevertheless, significant TFP puncta persisted at these sites, which we interpret as SMAPs (Fig. 7a; Supplementary Movie 3). Interestingly, MCG fusion event generated multiple SMAPs that became more distinct over time (see framed inset Fig. 7a). To verify this finding we co-transfected wild-type CTLs with GzmB-mCherry and TSP-1-GFPspark and followed SMAP release by TIRF microscopy. We again observed the simultaneous release of co-localizing GzmB and TSP-1 in several, closely

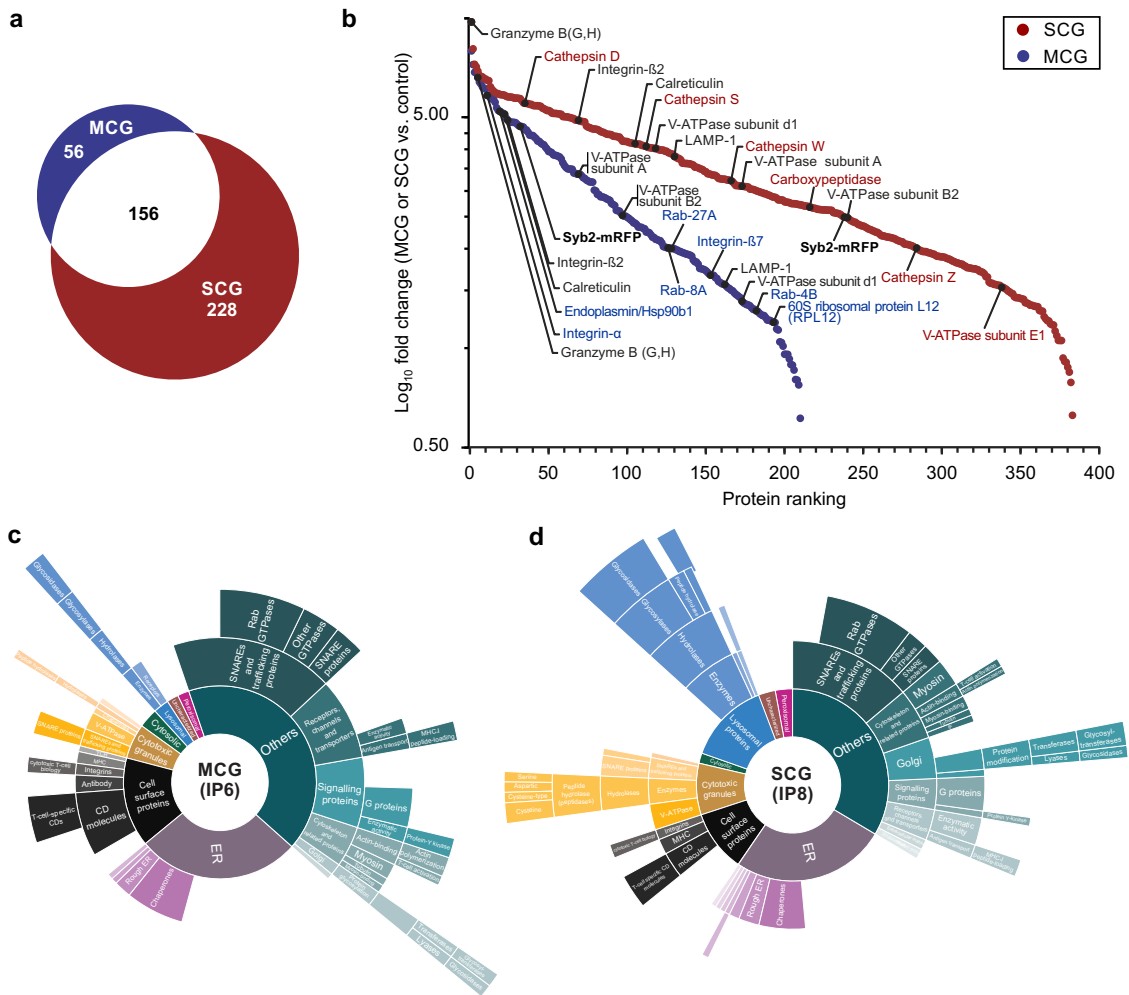

**Fig. 4 Multi core granules (MCG) and single core granules (SCG) exhibit a different protein composition as detected by mass spectrometry. a** Venn diagrams of all proteins identified from MCG and SCG. 56 proteins were unique in MCG and 228 proteins were unique in SCG, while 156 proteins were found in both classes. **b** Ranked abundance plot of significantly enriched proteins in the proteome of MCG and SCG. Significantly enriched proteins in MCG (blue dots) and SCG (red dots) compared to the naive cell control plotted as $Log_{10}$ fold change against the ranked protein abundance. Proteins specifically found in MCG and SCG are shown in blue and red, respectively. The common proteins are labeled in black. **c, d** Gene ontology analysis and manual annotation of enriched gene ontology terms. Annotations (cellular localization and biological function) assigned to the enriched gene ontology terms in MCG (**c**) and SCG (**d**) were conducted manually in agreement with current literature and database evidence. The inner core segments (e.g., lysosomal proteins, endoplasmic reticulum, cytosolic, etc.) of the sunburst plots represent the annotations based on cellular localization of the proteins. Of note, the size of each segment corresponds to the number of genes associated with this annotation term. Outer segments indicate biological functions of the proteins from the main categories. Five and three biological and technical replicates were performed for IP6 and IP8, respectively.

clustered puncta (Fig. 7d, Supplementary Movie 4). The most likely explanation for this phenomenon would be the release of several SMAPs from a single MCG, which then remain attached at the SLB membrane as individual entities. We further assessed the cytotoxic effect of MCGs and SCGs on target cells. For this we centrifuged for 1 h either fraction 6 or 8 down on poly-ornithine coated glass coverslip at high speed to partially crack open the granules. Then we seeded P815 target cells on them and at indicated times we fixed and immuno-stained them with anti-cleaved caspase3 antibody, which is an early marker for apoptosis. Already after 6 h we observed that either SCG or MCG containing fractions were able to induce cleaved caspase3 expression in target cells (Fig. 7e, f). In long term (19 h), MCGs were more efficient to kill target cells as SCGs under our experimental conditions, most likely due to the concentrated amount of cytotoxic proteins stored in SMAPs (Fig. 6b). Hence, we clearly demonstrated that SMAPs are being stored in cytotoxic MCGs that are released upon IS formation.

## Discussion

We have isolated fusion-competent Syb2 positive granules from mouse CTLs by ultracentrifugation and immuno-isolation and performed a detailed analysis of the CGs by mass spectrometry, super-resolution microscopy, live TIRF microscopy and electron microscopy. Besides the expected SCG we now report a distinctive class of fusion-competent granules with distinct morphology and protein composition, which we refer to as MCG. Interestingly, we found similar classes of granules in human NK cell, which have potential use for cancer therapy[28] (Supplementary Fig. 4). MCGs contain distinct SMAPs and/or several bilayer membrane-based vesicles, and they fuse at the IS in parallel to SCG. Thus, we have identified the source of recently described SMAPs, opening avenues for future immuno-therapeutic approaches targeted at MCGs and optimization of NK- and CTL-mediated killing.

A wealth of studies over the past decades have investigated the mechanism by which CTLs kill their targets. The generally

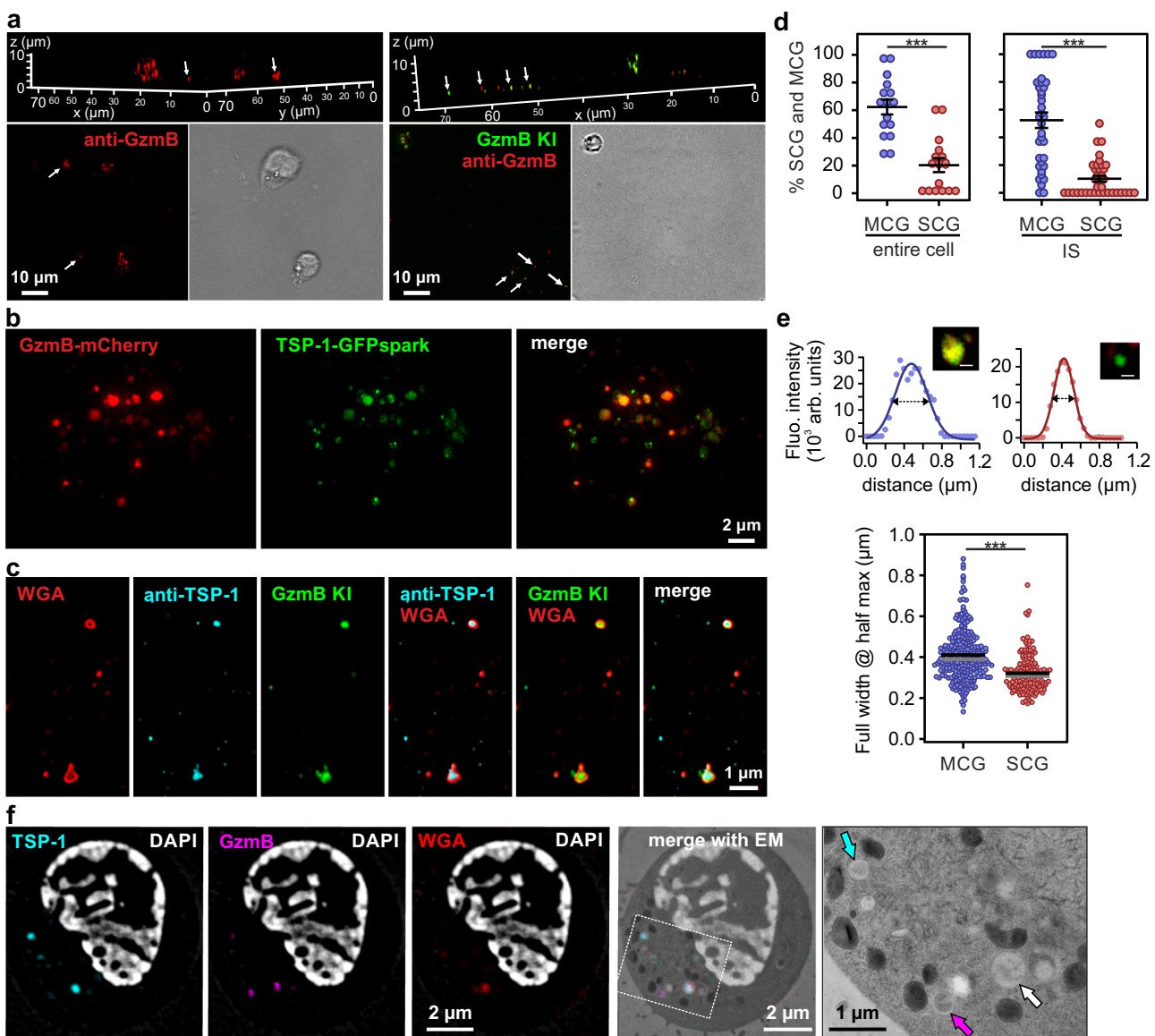

**Fig. 5 Multi core granules contain supramolecular attack particles (SMAPs). a** SIM images of WT (left) and GzmB KI (right) cells on supported lipid bilayers (SLB) represented in 3D (top) and 2D (bottom). Cells were stimulated on anti-CD3 antibody coated lipids for 90 min and stained with anti-GzmB (red) antibody after fixation. White arrows point to secreted SMAPs. $N_{mouse} = 1$; $n_{images} = 3$ and 6 for WT and GzmB KI, respectively. **b** SIM images of one resting WT CTL expressing GzmB-mCherry and Thrombospondin 1 (TSP-1)-GFPspark seeded on poly-ornithine coated coverslip. $N_{mouse} = 1$; $n_{cells} = 14$. **c** SIM images of secreted SMAPs on SLB from GzmB KI CTLs pre-incubated with wheat germ agglutinin (WGA)-647 to label intracellular, glycoprotein-rich compartments. Secreted SMAPs were stained with anti-TSP-1 antibody. $N_{mouse} = 1$, $n_{images} = 16$. **d** Analysis of SCG and MCG percentage in resting cells (left) and stimulated cells at the IS (right). To trigger synapse formation CTLs were seeded on anti-CD3 coated coverslips. GzmB single positive granules were identified as SCGs, while GzmB and WGA double positive granules as MCGs. Black lines represent mean ± sem; $n_{cells} = 16$; entire cells and $n_{cells} = 35$; IS. Student's t-test was used to compare values; *** Two-tailed P-value < 0.001. **e** Line profile (top) of one representative MCG (left) and SCG (right) acquired with SIM to depict granule diameter measurement method. Inset: SCG contained only GzmB (green) while the MCG contained WGA-647 and GzmB (yellow). Scale bar, 0.2 μm. Bottom scatter dot plot of granule diameter measured as the full width at half maximum in the line plots. Black line represents mean and gray line median. $N_{mice} = 3$; $n_{cells} = 20$; $n_{granules} = 275$ and 156 for MCG and SCG, respectively. Mann–Whitney U test was used to compare values; ***p < 0.001. **f** CLEM images of a representative stimulated CTL as in (**b**) pre-incubated with WGA-647 and stained with DAPI. Shown are SIM images and its corresponding TEM overlaid image. In the enlarged TEM image, arrows point to different fluorescent proteins using the color-code of the upper panels. White arrow indicate the MCG marked in all three channels. $N_{mouse} = 1$; $n_{cells} = 18$. Source data are provided as a Source data file.

accepted picture is that upon T cell receptor mediated recognition CGs are being brought to the target cell interface, where they fuse and release their toxic content. Death is caused by two pathways, the Prf1/Gzm pathway and the Fas/Fas-ligand pathway, which both induce apoptosis and/or necrosis in the target cell. Because proteins from both pathways have been found on classical CGs with a single dense core, it was assumed that exocytosis of one

class of CGs is responsible for target cell death. That simplistic view has been challenged by CG composition analysis[29] and the report of cytotoxic exosomes released by NK cells, and CTLs[30–33]. Bulk analysis of these exosome preparations suggested that FasL, GzmB, and Prf1 were present along with tetraspanins [34]. These particles dispayed cytotoxic activity against tumors in vivo[30,32,33]. An alternative view of cytotoxic exosomes is that exosome

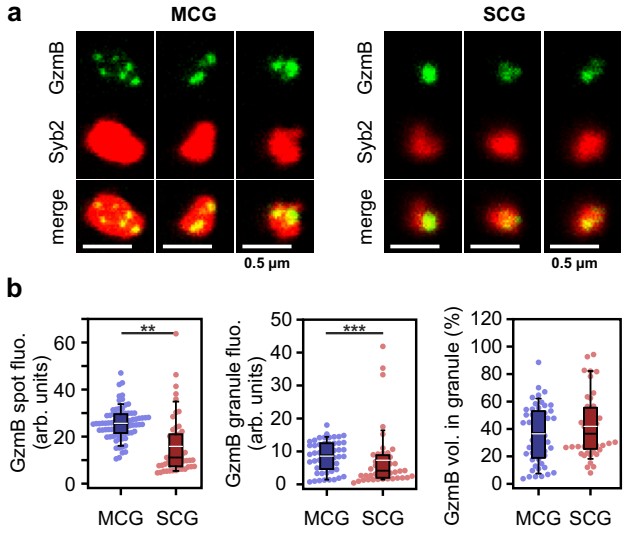

**Fig. 6 MCG contain more GzmB than SCG. a** 2DSTED single plane images of individual granules from fraction 6 (left) and 8 (right), that were isolated from Syb2 KI mice, and were gently seeded on glass coverslips to ensure their integrity. Granules were stained with anti-GzmB (top, green) and anti-RFP with STED compatible secondary antibody to visualize Syb2 (middle, red). The bottom images show the merged channels. No deconvolution was applied on the images in order to preserve the diffuse GzmB staining visible especially in SCGs. **b** Analysis of GzmB content of the granules depicted in (**a**) shown as scatter dot plots superimposed with a box plot, in which the center black line is the median and the white line the average. The boundaries of the box represent the 25th to 75th percentiles, while the whiskers are 10th/90th percentiles. Left, mean fluorescence intensity of the GzmB particles. Center, mean fluorescence of GzmB spots related to the entire volume of the granule measured on the Syb2 channel. Right, fractional GzmB volume over the entire volume the granule given in percent. $N_{gradient} = 1$; $n_{granules} = 52$ and 44 from fraction 6 (MCGs) and 8 (SCGs) respectively. $**p = 0.006$, $***p < 0.001$ computed with the Mann–Whitney $U$ test. Source data are provided as a Source data file.

fractions from NK cells contain FasL-positive exosomes, but that Prf1 and GzmB may be contained in co-purified SMAPs due to their similar size and density[22,24]. SMAPs consist of a core of GzmB and Prf1 surrounded by a glycoprotein shell, and lack a limiting lipid bilayer[22]. They were also shown to be able to kill target cells, and again no SMAPs have been found within single dense core granules. The new MCGs we identify in this work now resolves these issues and most likely represent the source of exosomes identified through 14-3-3ε[35], and SMAPs containing Prf1 and GzmB.

We envision MCGs as multi-purpose organelles containing SMAPs, exosomes and probably also cytokines that can fuse in parallel with SCGs at the IS (Fig. 8). MCGs as well as SCGs can kill target cells (Fig. 7e, f), and MCGs appear to contain more GzmB than SCGs (Fig. 6b). It is tempting to speculate that SCGs and MCGs might function under different processes in the attack of CTLs against target cells. While SCGs might act to rapidly kill a relatively small number of target cells, MCGs secretion gains relevance as a second line of attack for targets that resist the soluble cytotoxic proteins. The extra release of packaged lytic molecules might then have a better chance to deliver sufficient Gzms to the cytoplasm of the target cell. In addition to cytolytic killing, GzmB has been shown to play an important function in CTL transmigration through parenchymal tissue[36]. It would be interesting to further explore whether this function is also supported by GzmB released from MCGs. Furthermore, the observed heterogeneity in MCG morphology and composition

(Figs. 3 and 4) may enable CTLs to further fine-tune their killing efficiency depending on which individual MCG is exocytosed (see Fig. 8 right for an example). Thus, the co-existence of functionally different lytic granules in individual CTLs might contribute to the heterogeneity of CTL responses and, in turn, to the robustness of their effector function[37–39]. In this context it will be interesting to investigate the mechanisms of MCG biogenesis and fusion in the future. While our data clearly show that fusion of MCG is a SNARE-dependent process[8], the efficiency of fusion might be regulated by intracellular factors such as local calcium concentration, phosphorylation status of involved proteins or presence of facilitating factors such as Munc13-4. Concerning the biogenesis, the differences in protein composition between MCGs and SCGs (Fig. 4) indicate that both granules have distinct maturation routes leading to two parallel secretory pathways. It will be interesting to study whether human genetic defects like Chediak-Higashi[40,41] and Hermansky-Pudlak[42] syndromes, which were instrumental in understanding the biogenesis of SCGs, also affect the biogenesis of MCGs.

At first glance the presence of two parallel classes of CGs used by CTLs for target cell killing might be surprising. However, neuronal synapses, which share many features with immunological synapses[43,44], have long been known to use two classes of vesicles for signaling[45,46]. Small clear vesicles (SCVs) contain the neurotransmitter molecules and mediate the fast response of postsynaptic neurons through postsynaptic potentials, while dense core vesicles (DCVs) contain peptides, hormones and growth factors, and signal on a prolonged time scale to post-synaptic neurons and adjacent glial cells. Interestingly, SCVs and DCVs not only have a different morphology and diameter, but also proceed through entirely independent pathways of biogenesis. However, they share identical SNARE proteins for their release. In addition, DCVs also fuse at cell somata and axons, while SCV exocytosis is restricted to active zones at the presynaptic terminal. Whether MCGs are released from non-synaptic sites in CTL should be investigated in the future.

In conclusion, we present evidence for the existence of a distinctive class of CGs that can be found in mouse and human cytolytic cells. Based on their morphological appearance, we name them MCGs. They are the source of SMAPs and represent a promising future target for modulating NK and T cell killing efficiency in immunotherapy. For this goal, future studies on MCG biogenesis, fusion and post-fusion signaling are needed.

## Methods

**Mice.** Synaptobrevin2-mRFP knock-in (Syb2 KI) and granzyme B-mTFP knock-in (GzmB KI) mice were generated as described previously[8,25]. Syb2 KI and GzmB KI crossbred mice (Syb2/GzmB DKI) were used in experiment shown in Fig. 2. Transgenic mice and wildtype mice used in this study were all in C57BL/6N background. Additionally, all the animals used for experiments were at the age of 15–22 week-old and of both sexes. Animals were kept under the housing conditions of 22 °C room temperature with 50–60% humidity and 12 h dark/light cycles. All experimental procedures were approved and performed according to the regulations by the state of Saarland (Landesamt für Verbraucherschutz, AZ.: 2.4.1.1).

**Cell culture.** Splenocytes were isolated from 15–22 week-old mice as described before[47]. Briefly, naive CD8 T cells were positively isolated from splenocytes using Dynabeads FlowComp Mouse CD8+ kit (Invitrogen) as described by the manufacturer. The isolated naive CD8+ T cells were stimulated with anti-CD3/anti-CD28 activator beads (1:0.6 ratio) and cultured for 3 days at 37 °C with 5% $CO_2$. Cells were cultured at a density of ~$13 \times 10^6$ cells/ml in T75 culture flasks with IMDM medium (Invitrogen) containing 10% FCS, 50 U/ml penicillin, 50 μg/ml streptomycin (Invitrogen), 30 U/ml recombinant IL-2 (Gibco) and 50 μM 2-mercaptoethanol. These activated effector CTLs were used for organelle isolation and mass spectrometry analysis. Naive CD8+ T cells were collected immediately after CD8 positive cell isolation from splenocytes without any stimulation as a control. For functional assay shown in Fig. 5 and Fig. 7, effector cells were cultured 5 days before the experiments. P815 target cells (ATCC TIB64, directly purchased from DSMZ; #ACC1) were cultured in RPMI medium (Invitrogen) containing 10%

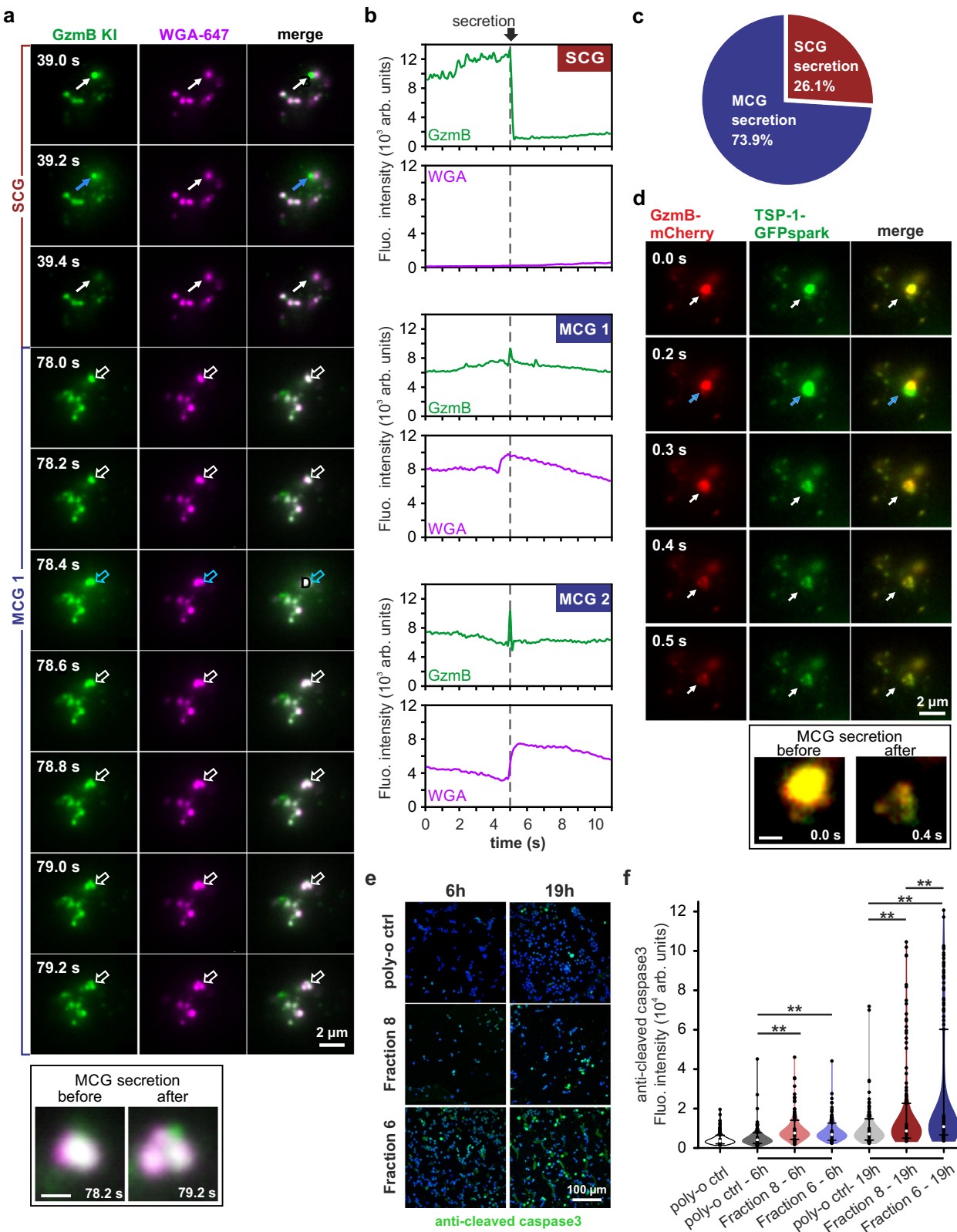

FCS, 50 U/ml penicillin, 50ug/ml streptomycin (Invitrogen), and 10 mM HEPES (Invitrogen) at 37 °C with 5% $CO_2$.

NK92 cell line (ATCC CRL-2407, kind gift from Prof. Borrow, University Oxford) were cultured in complete NK cell medium ((RPMI-1640 (Sigma Aldrich; #R8758), 10% FBS (Thermo Fisher Scientific; #A3160801), 100 U/ml penicillin, 100 μg/ml streptomycin (Thermo Fisher Scientific; #15140122)) supplemented with 100 units/mL of recombinant human IL-2 (PeproTech; #200-02). Cells were cultured every three days and kept at 37 °C in humidified incubator with 5% $CO_2$.

**Plasmids and antibodies**. Human TSP1-GFPspark was cloned in the pMAX vector by digesting the Thrombospondin-1/TSP1 cDNA ORF Clone, Human, C-GFPSpark tag from SinoBiological (cat # HG10508-ACG) with KpnI and XbaI FastDigest Restriction Enzymes (Thermo Scientific). The same was performed for pMAX vector, and both the vector and insert were ligated using T4 DNA ligase (Promega). The GzmB-mCherry construct was generated by fusing mCherry at C-terminus of mouse *Gzmb* gene. Additional GGSGGSGGS linker sequence was inserted between GzmB and mCherry. GzmB-linker-mCherry was then cloned into

**Fig. 7 Both classes of CGs, MCG and SCG, fuse at the IS and are killing competent. a** Total internal reflection fluorescence microscopy (TIRFM) snapshot images of GzmB KI (green) CTL stained with WGA-647 (magenta) on supported lipid bilayer. White solid arrows point to one SCG fusion event and white open arrows point to one MCG fusion event over time. Blue arrows indicate the time of fusion. Pictures in frame show the MCG prior to fusion (before) and releasing several SMAPs (after). Note Images were recorded at 5 Hz. $N_{mice} = 3$, $n_{cells} = 16$. **b** Representative granule secretion profile of SCG and MCG from the cell shown in (**a**). The loss of fluorescence indicates the release of soluble GzmB and the remaining/increased WGA signal indicates the secreted SMAPs on the supported lipid bilayer. **c** Percent of secreted MCG and SCG from experiments as shown in (**a**); $N_{mice} = 3$, $n_{cells} = 25$. **d** TIRFM snapshot images of a WT CTL expressing GzmB-mCherry (red) and TSP-1-GFPspark (green) on glass coverslip coated with an anti-CD3 antibody. White arrows point to one MCG secretion event, blue arrows indicate the time of secretion. Frame shows the MCG before and after fusion. Images were recorded at 10 Hz. $N_{mice} = 2$, $n_{cells} = 17$. **e** Target cell killing assay. Confocal images of P815 target cells seeded coverslips covered with MCG (fraction 6) and SCG (fraction 8). Cells were stained with anti-cleaved caspase3 antibody (green) to determine cell apoptosis at two different time points. Cells seeded on poly-ornithine alone were used as control. DAPI staining (blue) was used to count the cells. **f** Time dependent analysis of the killing assay is shown as a violin plot with superimposed box plot and outliers. The boundaries of the box represent the 25th to 75th percentiles, while the whiskers are 10th/90th percentiles. The white dot corresponds to the median. $N_{mouse} = 1$, $n_{cells} = 300$. N, n correspond to the number of independent experiments and analyzed cells, respectively. One-way ANOVA on rank was used to compare values within time point group; ***$p < 0.001$. Source data are provided as a Source data file. Scale bar in framed pictures are 0.5 μm.

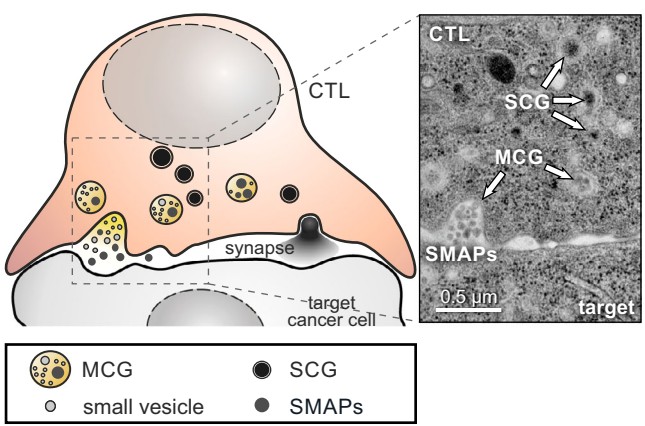

**Fig. 8 MCG as a distinct class of cytotoxic granules for SMAP release.** Proposed model of a cytotoxic T lymphocytes that contains two classes of CGs (MCG and SCG), both of which can fuse in parallel at the immunological synapse (left). While the SCG content diffuses rapidly after release, MCGs deposit supramolecular attack particles (SMAPs) that retain their integrity in the synaptic cleft. The latter is illustrated in the exemplary electron micrograph on the right.

pMAX vector[25]. The antibodies used in this work are described in detail in Table 1. All the antibodies except anti-Sec61β antibody are commercially available and are well-established and shown to be valid. The validation of the in-house generated antibody anti-Sec61β antibody is shown in the Source Data file (Supplementary Fig. 2a).

**Flow cytometry**. For flow cytometry analysis, $0.2–0.5 \times 10^6$ CD8+ T cells were resuspended in D-PBS (Gibco) and incubated in the dark for 30 min on ice with cell surface specific FITC-, APC or PE-conjugated antibodies against CD44, CD62L and CD25 (Table 1). The viable T lymphocytes was gated based on their size and granularity. Effector Syb2 KI cells were gated using the mRFP signal compared to the baseline signal of wild type cells. The fluorescence signal of antibody stained cells were further analyzed by subtracting mRFP signal in unstained Syb2 KI cells. Finally, the immune cell subsets were analyzed according to the antibody labeling. Data were acquired by using a BD FACSAria III analyzer (BD Biosciences) with BD FACSDiva software 6.0. The data were analyzed with FlowJo v10.0.7 software.

**Cell homogenization and subcellular fractionation**. $0.8-1.2 \times 10^8$ activated CD8+ T cells from one Syb2 KI mouse, $0.7–0.8 \times 10^8$ unstimulated naive CD8+ T cells from four to six Syb2 KI mice or $0.8 \times 10^8$ activated CD8+ T cells from one Syb2 KI/GzmB DKI mouse were harvested and washed once in buffer (Invitrogen, PBS with 0.1% BSA and 2 mM EDTA) before resuspension in 2 ml of homogenization buffer (300 mM sucrose, 10 mM HEPES (pH 7.3), 5 mM EDTA (pH 8.0) supplemented with protease inhibitors (3 mM Pefabloc, 10 μM E64 and 10 μg Pepstatin A). The cell suspension was transferred into a pre-chilled cell disruption bomb (Parr 1019HC T304 SS) connected to a nitrogen source. After 25 min equilibration at 800 psi the nitrogen pressure was released and the cell homogenate was collected. The cell lysate was centrifuged for 10 min at $1000 \times g$ at 4 °C to pellet unbroken

cells, partially disrupted cells and nuclei. The resulting post-nuclear supernatant was layered on top of a discontinuous sucrose density gradient with 0.8, 1.0, 1.2, 1.4, and 1.6 M sucrose in 10 mM HEPES with 5 mM EDTA with protease inhibitors as described before (pH 7.3), 2 ml for each fraction. After ultracentrifugation at $100,000 \times g$ for 90 min at 4 °C in a SW40Ti rotor (Beckmann), twelve 1 ml fractions were collected from the top of the gradient and supplemented with fresh protease inhibitors. 61 μl of each fraction was added to 16 μl of 4× Lithium dodecyl sulfate (LDS) sample-buffer (Serva) and 50 mM DTT and heated at 96 °C for 7 min prior to SDS-PAGE and immunoblotting.

**Western blot**. Sucrose fractions were separated by SDS-PAGE on precast 10% Bis-Tris gels with MES running buffer or 4–12% gradient Bis-Tris gels with MOPS running buffer (NuPage, Invitrogen). Proteins were then transferred to 0.2 μm pore-size nitrocellulose membrane and blocked with 5% non-fat dry milk powder in TBS buffer containing 20 mM Tris, 0.15 M NaCl, and 0.05% Tween 20, pH 7.4 for 2 h at 20 ± 2 °C. For quantitative western blots protein concentrations in sucrose fractions were measured using Pierce660 reagent (Sigma Aldrich). The immunoblots were analyzed with anti-alpha 1 Na+/K+-ATPase, anti-SDHA, anti-alpha tubulin, anti-perforin, anti-β actin, anti-synaptobrevin2, anti-RFP, anti-granzyme B, anti-RPS10, anti-Sec61β, NFAT1 and horseradish peroxidase (HRP)-conjugated goat anti-mouse (H + L or light chain for immunoprecipitation) or anti-rabbit secondary antibodies (F(ab')2, heavy or light chain) (Table 1). For reprobing western blot membranes were stripped in stripping solution (Invitrogen) for 5–15 min at 20 ± 2 °C. Finally, the blot was developed using enhanced chemoluminescence reagents (SuperSignal West Dura Chemoluminescent Substrate; Thermo Fisher Scientific) and imaged by gel documentation (FluorChem M system, ProteinSimple).

**Immunoprecipitation**. Dynabeads Protein G magnetic beads (Invitrogen) were washed two times in PBS before use and 15 μl Dynabeads were incubated with 5 μg anti-Syb2 antibody (Table 1) and rotated for 40 min at 20 ± 2 °C. Syb2-mRFP positive fractions 6 and 8 were diluted 1:1 in 320 mM KCl, 10 mM HEPES, 5 mM EDTA solution (pH 7.3), added to the antibody-conjugated Dynabeads and rotated at 20 ± 2 °C overnight. After five washing steps with 500 μl 160 mM KCl buffer the Dynabeads were collected for western blot, LSM, TEM, SEM analysis or mass spectrometry. For western blot analysis 61 μl supernatant was mixed with 16 μl 2× LDS buffer and 50 mM DTT, heated at 96 °C for 7 min and stored at −20 °C. For mass spectrometry Dynabeads were mixed with 18 μl 2× LDS and 2 μl DTT (1 M), heated at 96 °C for 10 min and kept at −80 °C.

**Total internal reflection fluorescence (TIRF) microscopy**. The TIRFM setup (Visitron Systems GmbH) was based on an IX83 (Olympus) equipped with a UAPON100XOTIRF NA 1.49 objective (Olympus), solid-state excitation lasers at 488 nm, 561 nm and 647 nm, an iLAS2 illumination control system (Roper Scientific SAS), an evolve-EM 515 camera (Photometrics) and a filter cube containing Semrock (Rochester) FF444/520/590/Di01 dichroic and FF01-465/537/623 emission filters. The setup was controlled by Visiview software (version 4.0.0.11, Visitron GmbH). For TIRFM, day 5 bead activated CTLs isolated from GzmB KI or WT mice were used. 2 h before the experiment, beads and IL-2 were removed from CTLs. $3 \times 10^5$ cells were resuspended in 30 μl of extracellular buffer (10 mM glucose, 5 mM HEPES, 155 mM NaCl, 4.5 mM KCl, and 2 mM MgCl₂) and allowed to settle for 1–2 min on anti-CD3ε antibody (30 μg/ml, Table 1) coated coverslips or supported lipid bilayer. Cells were then perfused with extracellular buffer containing calcium (10 mM glucose, 5 mM HEPES, 140 mM NaCl, 4.5 mM KCl, 2 mM MgCl₂ and 10 mM CaCl₂) to stimulate CG secretion.

In order to distinguish SCG and MCG, cells were co-labeled with GzmB and SMAP markers, WGA or TSP-1. To label SMAPs in T cells, cells were either transfected with SMAP marker TSP-1-GFPspark construct or stained with

### Table 1 Antibodies used in this study.

| Antibody anti- | Supplier | Identifier | Source | Dilution | application |
|---|---|---|---|---|---|
| β-Actin | Sigma-Aldrich | AC-15 | Mouse | 1:5000 | WB |
| CD3ε | BD Pharmingen | 145-2C11 | Mouse | 3:100 | Glass coating |
| Cleaved Caspase-3 (Asp175) | Cell Signaling | 9661S | Rabbit | 1:200 | ICC |
| CD25-FITC | BD Pharmingen | 7D4 | Rat | 1:200 | FACS |
| CD44-APC | eBioscience | IM7 | Rat | 1:200 | FACS |
| CD62L-FITC | BD Pharmingen | MEL-14 | Rat | 1:200 | FACS |
| Granzyme B | Cell Signaling | 4275S | Rabbit | 1:5000 | WB |
| Granzyme B | Invitrogen | GB11 | Mouse | 1:200 | ICC |
| Granzyme B- Alexa647 | Biolegend | GB11 | Mouse | 1:100 | ICC |
| Mouse STAR red | abberior | STRED-1001 | Goat | 1:100 | ICC |
| α1 Na$^+$/K$^+$-ATPase | Abcam | 464.6 | mouse | 1:1000 | WB |
| NFAT1 | Cell Signaling | 4389S | Rabbit | 1:100 | WB |
| Perforin-1 | Invitrogen | PA1-22489 | Rabbit | 1:500 | WB |
| Rabbit STAR 580 | abberior | ST580-1002 | Goat | 1:100 | ICC |
| RFP | Genway Biotech | GWB-3BF-397 | Rabbit | 1:5000 | WB |
| RPS10 | Abcam | EPR8545 | Rabbit | 1:10,000 | WB |
| SDHA | Abcam | 2E3GC12FB2 AE2 | Mouse | 1:2000 | WB |
| Sec61β | HOMmade | human aa2-10 | Rabbit | 1:500 | WB |
| Synaptobrevin2 | Synaptic Systems | 69.1 | Mouse | 1:1000 | WB |
| Thrombospondin-1 | Cell Signaling | D7E5F | Rabbit | 1:100 | ICC |
| α-Tubulin | Abcam | ab4074 | Rabbit | 1:5000 | WB |
| Mouse IgG, (H + L), HRP | Invitrogen | 32430 | Goat | 1:1000 | WB |
| Mouse light chain, HRP | Merck | AP200P | Goat | 1:5000 | WB |
| Rabbit heavy chain HRP | Abcam | 2A9 | Mouse | 1:5000 | WB |
| Rabbit IgG, F(ab')2, HRP | Merck | AQ132P | Goat | 1:10,000 | WB |
| Rabbit light chain HRP | Merck | MAB201P | Mouse | 1:5000 | WB |

WGA-Alexa647 (Thermo Fisher Scientific, W32466). For WGA staining, GzmB KI cells were pre-incubated with 1 µg/ml WGA-Alexa647 for 1.5 h at 37 °C allowing WGA being endocytosed for secretion analysis. For another independent labeling, WT cells were co-transfected TSP-1-GFPspark and GzmB-mCherry plasmids. Cells were recorded for 7 min at 20 ± 2 °C. 488 nm, 561 nm, and 647 nm excitation laser were used to visualize mTFP/GFP, GzmB-mCherry and WGA-Alexa647. The images were taken with an acquisition frequency of 5 Hz and exposure time of 50 ms for GzmB KI and WGA recording and an acquisition frequency of 10 Hz and exposure time of 100 ms for TSP1-GFPspark and GzmB-mCherry recording. Images and time-lapse series were analyzed using ImageJ or the FIJI package of ImageJ. CG secretion analysis was performed using ImageJ with the plugin Time Series Analyzer. A change of fluorescence within 200 ms with accompanied diffusion cloud in GzmB channel was defined as fusion.

**Supported lipid bilayers (SLB).** SLB were prepared for TIRF imaging to visualize granule secretion at immunological synapse and secreted SMAPs attached to the SLB after degranulation. SLBs were prepared as previously described[48,49]. Briefly, to prepare a clean glass chamber for SLB, the glass coverslips pre-washed with acid piranha and a plasma cleaner were mounted on sticky-Slide VI0.4 (Ibidi) to form 6 flow channels. Small unilamellar liposomes were prepared using 18:1 DGS-NTA(Ni) (790404C-AVL, Avanti Polar Lipids), 18:1 Biotinyl Cap (870282C-AVL, Avanti Polar Lipids), and 18:1 (D9-Cis) PC (850375C-AVL, Avanti Polar Lipids) in specific mixtures at total lipid concentration of 4 mM. SLB were allowed to form by incubating 50 µl of liposome suspension per flow channel for 20 min at 20 ± 2 °C. SLB were then washed with HEPES buffer containing 1 mM CaCl$_2$ and 2 mM MgCl$_2$ (HBS) and blocked with 1% human serum albumin (HBS/HSA) prior to functionalization. The SLB were functionalized in two steps. First with 5 µg per channel streptavidin (S11226, Thermo Fisher Scientific) for 10 min at 20 ± 2 °C followed by three washings. Finally, 30 µg per channel biotinylated anti-mouse CD3ε (BD Pharmingen, clone 145-2C11) was linked to the streptavidin on SLB as well as 50 µg/ml 12-Histidine tagged mouse ICAM-1 was linked to Nickel ions.

**Confocal imaging.** The killing of target cells by CTLs was visualized by confocal microscopy (LSM 780, Zeiss). Briefly, day 5 effector Syb2 KI CTLs (1 × 10$^5$) were co-cultured with P815 target cells (2 × 10$^4$) on poly-ornithine coated glass coverslip to ensure less cell movement for imaging. The IMDM co-culture medium contained 10 mM HEPES and 10 µg/ml anti-CD3ε antibody (Table 1) to promote T cell to form contacts with target cells. Live imaging was performed under 37 °C temperature control. Images were acquired as z-stacks over time. The total thickness of the stack was 8 µm while distance between individual slices was 1 µm.

To evaluate pH value in MCG, day 4 GzmB KI CTLs were first loaded with 2.5 µg/ml WGA-Alexa647 for 30 min and added additional 100 nM LysoTracker Red DND-99 (Thermo Fisher Scientific) in culture medium for another 1 h, allowing dyes to label MCG and acid compartment respectively. After staining, cells were washed twice with culture medium before imaging.

For fluorescence analysis of each sucrose fraction of Syb2 KI CTLs 1 ml was diluted in 10 ml D-PBS and centrifuged for 30 min at 10,000 × g at 4 °C in a SW41Ti rotor (Beckmann) on gelatin-coated coverslips[50]. For quality control 500 µl of sucrose fraction 6 and 8 of Syb2/GzmB DKI CTLs were pooled and centrifuged on coverslips for fluorescence analysis. The images were acquired with a 63x Plan-Apochromat objective (NA 1.4; Zeiss) with laser excitation at 561 nm for Syb2 KI signal and 488 nm for GzmB KI signal. The maximum projection images are shown. Object based co-localization analysis was performed with DiAna Plugin of imageJ [51]. Individual fluorescent spots were identified with the iterative segmentation procedure using the following settings: step value 100, size: 15-700 pixel, threshold was adjusted independently for each channel.

**Evaluation of MCG and SCG cytotoxicity.** MCGs (fraction 6) and SCGs (fraction 8) were isolated from sucrose gradient fractionation as described above. After dilution in 160 mM NaCl buffer with 10 mM HEPES (pH 7.3) another step of centrifugation was done for 60 min at 15,000 × g at 4 °C in a SW41Ti rotor (Beckmann) to settle down the organelles of fraction 6 and fraction 8 separately on poly-ornithine coated coverslips. Under these stronger centrifugation conditions, the organelles were partially disrupted to release their contents. Poly-ornithine solution was coated on 12.5 mm glass coverslips overnight at 4 °C for organelle immobilization. 1 × 10$^5$ P815 target cells in 100 µl were added to each organelle containing coverslip. Cells were incubated with granules for 6 h and 19 h at 37 °C. Finally, cells were fixed and stained with anti-cleaved caspase3 (Table 1) to evaluate cell death.

**Structured illumination microscopy (SIM).** T cells were fixed in ice-cold 2% PFA in Dulbecco's PBS (Thermo Fisher Scientific) for 20 min. For staining, cells were permeabilized with 0.1% TritonX-100 in Dulbecco's PBS (permeabilizing solution) for another 20 min followed by 30 min blocking in solution containing 2% BSA prepared in permeabilizing solution. Cells were stained with either Alexa 647 conjugated anti-GzmB (Biolegend) or anti-thrombospondin-1 antibodies (Table 1). Finally, cells were mounted with Mowiol based mounting medium and observed at SIM microscope (Zeiss Elyra PS.1).

Images for correlative fluorescence and electron microscopy (CLEM) of organelles bound to antibody-conjugated Dynabeads Protein G were acquired with excitation light of 488 and 561 nm wavelengths. Almost the entire field of view of a

200-mesh grid (around 90 μm²) could be observed with a ×63 Plan-Apochromat objective by SIM, allowing a perfect orientation relative to the grid bars. After adjusting the highest and lowest focus planes for z-stack analysis in brightfield, images were recorded with a step size of 100 nm to scan the organelle-Dynabead complexes.

Fluorescent images for CLEM of SMAP containing cells were excited with 405, 488, 561, and 647 nm wavelengths to visualize DAPI, TSP-1-GFPspark, GzmB-mCherry and WGA-647, respectively. The DAPI image was recorded to identify both the nucleus of the CTL and the image plane. 3–10 images were recorded with a step size of 100 nm to scan the cells of interest. All the images were acquired with a ×63 Plan-Apochromat (NA 1.4) and then processed to obtain higher resolutions by Zen 2012 (Zeiss).

**Stimulated emission depletion (STED) microscopy.** 500 μl of sucrose gradient fraction 6 and 8 generated from Syb2 KI CTLs were fixed with ice-cold 0.2% PFA for 10 min, diluted in 320 mM KCl buffer with 10 mM HEPES (pH 7.3) and centrifuged for 30 min at $10,000 \times g$ at 4 °C in a SW41Ti rotor (Beckmann) on gelatin-coated coverslips. For staining, organelles were permeabilized with 0.005% Triton X-100 in Dulbecco's PBS following blocking in PBS containing 2% BSA. Organelles were stained with anti-RFP and anti-GzmB antibodies, while the secondary antibodies were anti-rabbit-STAR580 and anti-mouse-STARred (Table 1). Finally, organelles were mounted with Abberior Mount solid antifade. Imaging was performed with a four-color STED QuadScan (Abberior) using 561 nm/2 mW, and 640 nm/12 mW excitation pulsed lasers, and a 775 nm/1.25 W STED laser. The pinhole size was set to 90 μm (1.13 arbitrary unit) and the probes were visualized with a ×100, NA 1.4 objective (UPLSAPO100XO, Olympus). The images were acquired using Inspector software V16.3 from Abberior. The following acquisition protocol was used. First, a single confocal section was recorded at 561 nm, and 640 nm to generate an overview of the granules. Then, single granules, in which both signals co-localized, were recorded in 2D STED mode generating a stack of 10 images with a total depth of 1 μm. Syb-2-mRFP and GzmB were visualized at 561 nm and 647 nm, respectively, with laser power set to 40% and 30%. The STED laser emitted 50% of the maximal power of 1250 mW (corresponding to 75–85 mW in the focus, repetition rate of 40 MHz) with a gating of 750 ps. The voxel size was $20 \times 20 \times 100$ nm. The percentage of GzmB in the granules was determined with DiAna Plugin[51] of imageJ. Individual fluorescent granules (561 nm channel) and GzmB particles (640 nm channel) were identified with the threshold segmentation procedure using the following settings: Gaussian filter: 0.5, spot size: 5 – ∞ pixel, threshold: 20 for the 561 channel and automatically set using Yen method for the 640 nm channel. The following parameters were analyzed: Volume, mean and integral brightness of the granules and GzmB particles. This allowed us to calculate 1. the mean fluorescence of GzmB per granule (Integrale GzmB fluorescence/Syb2-mRFP volume), which is related to the granule's GzmB concentration; 2. the percentage of the granule's volume that is occupied by all contained GzmB particles. Aside from background subtraction, the images were not processed in any way before analysis and for presentation purpose.

**Electron microscopy.** Sucrose fraction 6 and 8 were diluted in D-PBS to a final sucrose concentration of 0.8 M and 10 μl of each fraction were dropped on a pioloform-coated 200 mesh copper grid (Plano) and incubated for 30 min. After fixation with 2% paraformaldehyde and 1% glutaraldehyde, samples were contrasted with UranyLess (Electron Microscopy Sciences)[52]. Electron micrographs were obtained using a Tecnai G2 12 Biotwin (Thermo Fisher Scientific).

After immunoprecipitation the complex of organelle and Dynabeads Protein G of fraction 6 and 8 were resuspended in 2% gelatin in PBS-buffer, pipetted into membrane carriers (1.5 mm×0.1 mm) and vitrified in a high-pressure freezing system (EM PACT2; Leica). Freeze substitution and embedding in Epon was done as described previously[53]. All samples were processed in an automatic freeze-substitution apparatus (AFS2; Leica). In brief, all samples were transferred into the precooled (-130 °C) freeze-substitution chamber of the AFS2. The temperature was increased from −130 to −90 °C over 2 h. Cryo-substitution was performed with 2% osmium tetroxide in anhydrous acetone and 2% water. The temperature was increased linearly from −90 °C to −70 °C over 20 h, from −70 °C to −50 °C over 20 h, and from −50 °C to −10 °C over 5 h. After washing with anhydrous acetone, the organelle-Dynabeads Protein G complexes were embedded in Epon-812 (30% Epon/acetone for 15 min at −10 °C, 70% Epon/acetone for 1 h at −10 °C and pure Epon for 1 h at 20 °C; Electron Microscopy Sciences). The temperature was increased linearly from 20 to 60 °C for 4 h, and epon was polymerized at 60 °C for 1 d. After polymerization, the membrane carriers were removed from the Epon block. Ultrathin (70 nm) sections were cut using an ultramicrotome (EM UC7; Leica), collected on Pioloform-coated copper grids, stained with uranyl acetate and lead citrate and analyzed with a Tecnai G2 Biotwin electron microscope (Thermo Fisher Scientific). Only organelles with well-conserved and intact membranes were analyzed in a close distance to the Dynabead surface. The transmission electron microscopy (TEM) images were acquired using Olympus iTEM 5.0 image software (build1243).

Correlative light-electron microscopy (CLEM) analysis of cryo-fixed organelle-Dynabead Protein G complexes of fraction 6 and 8 was done as described previously[8]. The membrane carriers with the frozen specimens were cryo-transferred into the precooled (−130 °C) freeze substitution chamber of the AFS2.

The temperature was increased from −130 to −90 °C for 2 h. Cryo-substitution was performed at −90 °C to −70 °C for 20 h in anhydrous acetone and at −70 to −60 °C for 24 h with 0.3% (wt/vol) uranyl acetate in anhydrous acetone. At −60 °C the samples were infiltrated with increasing concentrations (30, 60, and 100%; 1 h each) of Lowicryl (3:1 K11M/HM20 mixture; Electron Microscopy Sciences). After 5 h of 100% Lowicryl infiltration, samples were UV polymerized at −60 °C for 24 h and for an additional 15 h while temperature was raised linearly to 5 °C. Samples were kept in the dark at 4 °C until further processing. After removing the membrane carriers, 100 nm ultrathin sections were cut using an EM UC7 (Leica) and collected on carbon-coated 200 mesh copper grids (Plano). Fluorescence analysis of EM grids was performed within 1 d after sectioning to avoid loss of fluorescence signals in the sections. For fluorescence microscopy the grids were placed on a drop of water between two coverslips and sealed with silicone (Picodent Twinsil). Sections were imaged with high resolution SIM. The very same grids previously analyzed with high-resolution SIM were stained with uranyl acetate and lead citrate and analyzed with the Tecnai 12 Biotwin electron microscope. For correlation the autofluorescent pattern of the Dynabeads and the grid orientation, observed in brightfield, was used to find the optimal overlay on the EM images. Images were overlaid in Corel DRAW X6.

For correlative light and electron microscopy (CLEM) analysis of SMAPs in CTLs, day 4 WT cells were used. CTLs were co-transfected with TSP-1-GFPspark and GzmB-mCherry for 16 h. To label SMAPs in T cells, transfected cells were pre-incubated with WGA-Alexa647 for 1.5 h at 37 °C allowing WGA being endocytosed. After incubation with WGA-Alexa647 4000 CTL were seeded on poly-L-ornithine (0.1 mg/ml) and anti-CD3ε (30 μg/ml) coated 1.4 mm sapphire discs in flat specimen carriers (Leica) and incubated for 25 min at 37 °C with 5% CO₂. Cells were vitrified in a high-pressure freezing system (EM PACT2, Leica) in AIMV medium containing 30% FCS. The vitrified samples were further processed for CLEM analysis as described above. After removing the sapphire discs ultrathin sections (100 nm) were cut using a Leica EM UC7 (Leica) and collected on carbon coated 200 mesh copper grids (Plano). 1 d after sectioning the grids were stained with DAPI for 3 min (1/10,000), washed and sealed between two coverslips for high resolution SIM imaging. After fluorescence imaging, the same grids were stained with uranyl acetate and lead citrate and recorded with the Tecnai12 Biotwin electron microscope. Only CTLs with well conserved membranes, cell organelles and nuclei were analyzed and used for correlation. For correlation, the DAPI 405 nm image showing the labeled nucleus of the cell was used to find the best overlap with the electron microscope image. The final alignment defines the position of the fluorescent signals within the cell of interest. Images were overlaid in Corel DRAW X6.

For scanning electron microscopy (SEM), immuno-isolated Syb2-positive granules attached to antibody-conjugated Dynabeads Protein G were dropped on a glass coverslip and fixed in 2% paraformaldehyde for 10 min at 20 ± 2 °C. The beads were washed in PBS and fixed with 1% glutaraldehyde. After several wash steps with water the beads were contrasted with 2% aqueous uranyl acetate solution. After graded dehydrated series of ethanol (30%, 50%, 60%, 70%, 80%, 90%, 100%, 100% for 2 min each) and drying with 50% and 100% for 2 min each with hexamethyldisilazane (HMDS, Sigma Aldrich)[54] the samples were sputter-coated with a layer of gold and analyzed with an XL30 ESEMFEG scanning electron microscope (FEI; Thermo Fisher Scientific) at 10 kV accelerating voltage in secondary electron (SE)- and backscattered electron (BSE)-modes. The SEM images were acquired using microscope control software (version 7.0).

**Cryo-soft X-ray tomography (CSXT).** NK-92 Cells were plunge frozen on carbon coated transmission electron microscopy (TEM) grids (Quantifoil, TAAB Laboratories equipment Ltd, Reading, UK; #G255). Tilt series were collected on the Xradia UltraXRM-S220c X-ray microscope (Zeiss) at the B24 beamline of the Diamond synchrotron with a Pixis-XO:1024B CCD camera (Teledyne Princeton Instruments, Birmingham, UK) and a 40 nm zone plate with X-rays of 500 eV. Tilt series were collected from −60° to +60° with an increment of 0.5°. X-ray tomograms were reconstructed using etomo, part of the IMOD package[55]. Post processing was performed with ImageJ (National Institute of Health).

**Imaging data analysis and statistical analysis.** All images were analyzed with image J or FIJI version 15 and above (National Institute of Health). Granule sizes and their contents from different imaging techniques were analyzed with Igor (Igor Pro 6.04) or SigmaPlot 14.0. All statistical tests were performed with Igor or SigmaPlot and data are represented as mean ± SEM (or SD when specified). The statistical tests are indicated in each figure legend. Unless mentioned otherwise, Mann–Whitney $U$ test was used for two-group comparison. All $p$-values were calculated with two-tailed statistical tests and 95% confidence intervals. **$p < 0.01$, ***$p < 0.001$. In the box plots, the boundaries of the box represent the 25th to 75th percentile, a black line within the box marks the median, while the white line corresponds to the mean. Whiskers above and below the box indicate the 10th and 90th percentiles, respectively.

**Sample preparation of immuno-isolated vesicles for mass spectrometry-based proteomics.** Proteins of immuno-isolated samples were resolved on NuPAGE 4–12% Bis-Tris gradient gels (Thermo Fisher Scientific), stained with

InstantBlue Coomassie Protein Stain (Expedeon) and subjected to in-gel digestion with trypsin as described [56]. Briefly, entire gel lanes were cut into 23 pieces of 1 mm$^2$, proteins were reduced with 10 mM DTT, alkylated with 55 mM iodoacetamide and digested with trypsin (Promega, V5111) overnight. Tryptic peptides were extracted, dried and reconstituted in solution containing 2% [v/v] acetonitrile and 0.05% [v/v] trifluoroacetic acid and subjected to liquid chromatography-tandem mass spectrometry (LC-MS/MS).

**LC-MS/MS analysis.** Peptides derived from three (IP8) and five (IP6) biological replicates were analyzed as technical duplicates on Q Exactive HF hybrid quadrupole-Orbitrap mass spectrometer (Thermo Scientific), coupled to a Dionex UltiMate 3000 UHPLC system (Thermo Scientific) equipped with an in-house-packed C18 column (ReproSil-Pur 120 C18-AQ, 1.9 μm pore size, 75 μm inner diameter, 30 cm length, Dr. Maisch GmbH). Samples were separated applying the following 48 min gradient: mobile phase A consisted of 0.1% formic acid [v/v], mobile phase B of 80% acetonitrile/0.08% formic acid [v/v]. The gradient started at 5% B, increasing to 10% B within 3 min, followed by 10–45% of B within 33 min, then keeping mobile phase B constant at 90% for 6 min. After each gradient, the column was again equilibrated to 5% B for 6 min. The flow rate was set to 300 nl/min. Eluting peptides were analyzed in positive mode using a data-dependent top 30-acquisition methods. MS1 spectra were acquired with a resolution of 60,000 in the Orbitrap covering a mass range of 350–1600 $m/z$. Injection time was set to 50 ms and automatic gain control (AGC) target to $1 \times 10^6$. Dynamic exclusion covered 25 s. Precursor ion charge state screening was enabled, all unassigned charge states were rejected and only precursors with a charge state of 2–7 were included. MS2 spectra were recorded with a resolution of 15,000 in the Orbitrap, injection time was set to 60 ms, AGC target to $1 \times 10^5$ and the isolation window to 1.6 $m/z$. Fragmentation was enforced by higher-energy collisional dissociation (HCD) at 30%.

**Mass spectrometry data analysis and visualization.** Raw files were processed by MaxQuant (MQ) software (version 1.6.0.1)[57,58] and its built-in Andromeda peptide search engine[59] with the following settings to identify proteins: trypsin/P was used as digestion enzyme with maximal two missed cleavage sites; tandem mass spectrometry (MS/MS) spectra were searched against a customized version of the April 2016 release of the UniProt complete *Mus musculus* proteome sequence database, in which the sequence of synaptobrevin2 (VAMP2) was completely replaced with sequence of C-terminally mRFP-tagged Syb2 protein; carbamidomethylated cysteines were set as fixed and oxidation of methionine as variable modification (maximal allowed number of modifications per peptide was set to five); maximum false discovery rate was set to 0.01 both on peptide and protein levels. Further parameters were used as set by the default settings.

MaxQuant's label-free quantification (LFQ) algorithm (MaxLFQ) was applied to calculate a data matrix of LFQ intensities that allowed the relative comparison of protein amounts across samples[60]. The default settings (MaxQuant version 1.6.0.1) for LFQ intensity calculations were applied, LFQ minimum ratio count of two for each pairwise comparison step was required, as well as unique and razor peptides were considered for the quantification.

To interpret the protein quantitation and co-enrichment data, analysis was performed by using the software platform Perseus (version 1.6.2.2)[61]. ProteinGroups.txt file and the corresponding intensity matrices (LFQ intensities) were loaded into Perseus and the results cleaned for reverse hits, contaminants and identified only by site. Positive intensity values were logarithmized (log$_2$). Using the "categorical annotation rows" option, biological and technical replicates for the corresponding sample pairs (affinity purified gradient fraction 6 and 8 from stimulated and naive (control) CTL) were set equal, (three and five biological replicates for fractions 8 and 6, respectively, were each measured as technical duplicates). Only proteins identified in at least 50 per cent of all replicates (valid value filter set to 50% in at least one group) were taken under consideration for the significance analysis. Signals, that were originally zero prior to conversion to log$_2$, were imputed with random numbers from a normal distribution (total matrix mode), whose mean and standard deviation were chosen to simulate low abundance values (width = 0.3; shift = 1.8). Significantly co-enriched proteins and interactors were determined by a volcano plot-based approach combining two sample $t$ test-derived $p$ values with (fraction # vs control) ratio information.

Significance lines in the volcano plot corresponding to a given FDR were determined by a permutation-based method [62]. Threshold (= FDR) and SO (=curve bend) values were set equal to 0.05 (5%) and 0.1, respectively. The resulting data tables were exported and subjected to further analysis in Excel 2016 (Microsoft Office Package 2016). Categorical annotation by biological function and/or subcellular localization was performed manually for all proteins that were identified as significantly enriched in each of the two Syb2 positive fractions. Protein annotations were summarized in annotation matrix tables with Excel 2016. Each annotation unit was based only on published scientific literature available on NCBI's website (National Center for Biotechnology Information, https://www.ncbi.nlm.nih.gov/) and UniProt (https://www.uniprot.org/)[63]. For the categorical annotation of proteins with enzymatic function, the enzyme nomenclature database provided by Nomenclature Committee of the International Union of Biochemistry and Molecular Biology (NC-IUBMB) was used as a source (https://www.qmul.ac.uk/sbcs/iubmb/enzyme/). Sunburst diagrams were created by

Excel 2016 software (Microsoft Office). The diagrams represented hierarchically the correlation between numeric and categorical variables generated during the protein annotations. To modify graphically the color representation and to add categorical labels, sunburst diagrams were imported as vector files into Adobe Illustrator CS5. Percentage of proteins in the main categories were calculated based on the total number of significantly enriched proteins for each fraction sample. Proportional Venn diagrams were generated using BioVenn web-based application[64]. Color adaptations of the BioVenn diagrams were undertaken using Adobe Illustrator CS5.

**Reporting summary**. Further information on research design is available in the Nature Research Reporting Summary linked to this article.

## Data availability

Mass spectrometry data are available in Supplementary Data 1. The mass spectrometry raw and MaxQuant output files were deposited to the ProteomXchange Consortium (www.proteomeXchange.org) via the PRIDE[65] partner repository with the dataset identifier PXD025055. Databases are available under the same dataset identifier. All original data sets (western blots, electron micrographs, CSXT images, immunofluorescent images (SIM, CLEM, and STED), Confocal and TIRFM movies) generated for this study and presented in the Source Data file have been deposited in ZENODO repository server (https://zenodo.org/) with the dataset identifier 10.5281/zenodo.5752116. The accession code will be provided by the corresponding author upon reasonable request. All other data are provided in the article and its supplementary files or from the corresponding author upon reasonable request. Source data are provided with this paper.

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

## Acknowledgements

This work was supported by grants from the Deutsche Forschungsgemeinschaft (SFB 894 (E.K., U.B., and J.R; SFB1286 to H.U. and R.J.), IRTG 1830 (J.R). Further, this project has received funding from the European Research Council (ERC) under the European Union's Horizon 2020 research and innovation program (grant agreement No 951329 to J.R., S.V., C.T.B., and M.L.D and No ERC-2014-AdG_670930 to S.B. and M.L.D.). Finally this work was also funded by the Kennedy Trust for Rheumatology Research (S.B. and M.L.D.) and Wellcome Trust for support of Diamond Light Source Ltd. We thank Margarete Klose, Anja Bergsträßer, Nicole Rothgerber, Anne Weinland, Nobert Pütz, Sabine König, Monika Raabe, Uwe Plessmann and Miroslav Gaydarski for excellent technical assistance. We thank Martin Jung for the anti-sec61 beta antibody and Abed Chouaib for ImageJ macro programming.

## Author contributions

Methodology and Investigation, H-F.C., C.S., M.N., U.H., U.B., S.B, M.H., and K.R.; Formal analysis, E.K. and U.B.; Resources and supervision, C.T.B., H.U., and R.J; Manuscript revision, H-F.C., C.S., U.B., S.V., C.T.B., and M.L.D.; Conceptualization, supervision, project administration, funding acquisition, writing original draft and editing, J.R.

## Funding

## Competing interests

The authors declare no competing interests.
