## [Peer Review File · Nature Communications]

Identification of distinct cytotoxic granules as the origin of supramolecular attack particles in T lymphocytesREVIEWER COMMENTS

Reviewer #1 (Remarks to the Author):

Chang et al. further embark on recent findings that synaptic secretion from cytotoxic lymphocytes is more complex than previously known and involves supramolecular attack particles (SMAPs) that are autonomously released from cytotoxic lymphocytes. SMAPs were shown to comprise large amounts of perforin and granzyme and may act outside of the immune synapse. Using a synaptobrevin2-mRFP knock-in mouse model in combination with immuno-isolation and mass spectrometry, the authors aimed to further analyse the heterogeneity of cytotoxic granules and identify the intracellular source for SMAPs. The authors were able to identify multiple core granules, in extension to previously known single core granules, that contain endosomal-like proteins. The authors found that both classes of cytotoxic granules fuse at the immunological synapse, but that SMAPs originate exclusively from the newly discovered multi core granula (MCG). The authors hypothesize that CTL use SCG fusion to fill the synaptic cleft with active cytotoxic proteins and in parallel MCG fusion to deliver latent SMAPs for delayed killing of refractory targets.

Given the very recent discovery of SMAPs and the importance of the understanding of the mechanisms by which cytotoxic lymphocytes kill, the above mentioned findings are of highest relevance and actuality with regard to cancer immunotherapies but also in the field of autoimmune diseases.

Minor remarks:

- The study analyses mouse CTL. As many differences between mouse and human immune cells are described in immunology it would be helpful to comment on the applicability of the findings to human human cells, especially in view of the extensive discussions with regard to the potential therapeutic translation.
- LAMP-1 (CD107a) is often used as degranulation marker. According to Figure 4 it seems that LAMP-1 is only present in SCG. Can one conclude that LAMP-1 can be used for visualization of SCG only but not MCG in FACS-based degranulation assays? Is there any good candidate for degranulation marker for MCGs?
- Lyotracker is also often used for live cell studies of lytic granules. It would very valuable to know if lysotracker labels both type of granules or only SCGs?
- Lytic granules are known to have very low pH. Did you compare pH in both types of SCG and MCG ?
- From the presented data, it seems that the majority of CGs in the cell are surprisingly MCGs and MCGs also represent the majority of secreted granules. What is the amount of Perforin and Granzyme B per granule volume in SCGs compared to MCGs?
- Could you comment on the contribution of both types of CGs to the killing capacity of the cell?

Torsten Tonn

Reviewer #2 (Remarks to the Author):

The study by Chang and colleagues identifies a novel cytotoxic granule component of CD8+ T cells that appears to be a major delivery mechanism of cytotoxic molecules. This is an impressive study that not only reconciles earlier descriptions of cytotoxic vesicles (SMAPs), but demonstrates that these granules (MCG vs SCGs) not only have distinct components but also likely form from distinct biogenesis. The observations in these studies will be well received by the field and open up new avenues of research by providing not only insights into the fact there are distinct classes of cytotoxic granules, but the different composition of these granules.

The manuscript is well written, with the conclusions well supported by the data and their interpretation. To that end, there are only a few issues that perhaps would help clarify some questions that came to mind during the review.

1. While the focus is on the role of cytolytic activity, there has been prior reports suggesting that cytolytic granules, particularly Granzyme B, have a role in CD8+ T cell trafficking through tissue (Prakesh et al., *Immunity*. 2014 Dec 18;41(6):960-72). Is there any indication that perhaps rather than both MCGs and SCGs having a role in cytolytic activity, there might be a role for either (or both) in transmigration through tissue based on protein composition?
2. While reassuring to see GrzB and Pfp present in these granules, there was an absence of other granzyme family members (eg GrzA, K, M). Is there a reason for this (type of CTL used)?
3. A quick question, Fraction 6 (MCG) is distinguished from Fraction 8 by Na⁺/K⁺ ATPase in figure 1. It was identified in the mass spec data but not highlighted in panel b, figure 4. Adding this might help with orientating readers.
4. TCR CD3 subunits, among other membrane proteins, are contained within MCGs. Is this due to the enveloped nature of some of the cores within organelle, or is there perhaps some regulatory function of these MCGs in regulating T cell activation?
5. While SMAPs have been described for NK cells, what isn't clear is whether NK cells have the same heterogeneity in cytolytic granules (MCGs vs SCGs)? Some discussion of this would be worthwhile.

Reviewer #3 (Remarks to the Author):

This paper describes the observation of two classes of fusion-competent granules, single core granules (SCGs) and multi core granules (MCGs), and defining their size, morphology and protein composition. In addition, functional analyses of these fusion-competent granules were conducted and it was shown that both classes of granules fuse with the plasma membrane at the IS. Thus a novel class of CGs named MCGs was discovered based on morphological appearance. MCGs could be a future target for modulating T cell killing efficiency in immunotherapy however this study was not conducted yet.

This paper, though describes important findings on mechanism by which CTLs kill their targets, would be a better fit for immunology related journals.

We thank the reviewers for their careful reviews, enthusiasm for our work, and constructive criticisms. We further thank the editors for the opportunity to revise the manuscript. We have performed new experiments and revised the text as outlined below in response to reviewer concerns and hope all agree that this has greatly improved the manuscript. All changes made in response to reviewer concerns are listed below and highlighted in the manuscript.

Reviewer #1

Chang et al. further embark on recent findings that synaptic secretion from cytotoxic lymphocytes is more complex than previously known and involves supramolecular attack particles (SMAPs) that are autonomously released from cytotoxic lymphocytes. SMAPs were shown to comprise large amounts of perforin and granzyme and may act outside of the immune synapse. Using a synaptobrevin2-mRFP knock-in mouse model in combination with immuno-isolation and mass spectrometry, the authors aimed to further analyse the heterogeneity of cytotoxic granules and identify the intracellular source for SMAPs. The authors were able to identify multiple core granules, in extension to previously known single core granules, that contain endosomal-like proteins. The authors found that both classes of cytotoxic granules fuse at the immunological synapse, but that SMAPs originate exclusively from the newly discovered multi core granula (MCG). The authors hypothesize that CTL use SCG fusion to fill the synaptic cleft with active cytotoxic proteins and in parallel MCG fusion to deliver latent SMAPs for delayed killing of refractory targets.

Given the very recent discovery of SMAPs and the importance of the understanding of the mechanisms by which cytotoxic lymphocytes kill, the above mentioned findings are of highest relevance and actuality with regard to cancer immunotherapies but also in the field of autoimmune diseases.

Minor remarks:

Comment 1: “The study analyses mouse CTL. As many differences between mouse and human immune cells are described in immunology it would be helpful to comment on the applicability of the findings to human cells, especially in view of the extensive discussions with regard to the potential therapeutic translation.”

Reply:

We appreciate the reviewer comment and agree that there are multiple studies showing differences between mouse and human immune cells. Therefore, to best address this point, we investigated whether the human NK cell line NK92, a potential immunotherapeutic candidate, also contains similar classes of organelles using Cryo-Soft-X-ray tomography. We found that NK92 cells contain two classes of organelles that share the same morphological features with mouse cells and had comparable sizes for SCGs and MCGs. We integrated these findings in the manuscript, page 6-7 Line 153 – 159 and as a new supplementary figure 4 and Supplementary Video 1 and 2. We expanded our material and method section accordingly (see page 14 and, Line 357 – 361 and page 24, Line 626 – 633). We also commented on these findings in the discussion page 11, line 273 - 278 (“Interestingly, we found ... mediated killing.”).

Comment 2: “LAMP-1 (CD107a) is often used as degranulation marker. According to Figure 4 it seems that LAMP-1 is only present in SCG. Can one conclude that LAMP-1 can be used for

visualization of SCG only but not MCG in FACS-based degranulation assays? Is there any good candidate for degranulation marker for MCGs?"

Reply:

The reviewer is right for pointing out that LAMP-1 (CD107a) was only shown for SCG in the protein ranking curve in Fig. 4b. and in the volcano plot Supplementary Fig. 4c. However, according to our mass spectrometry data, LAMP-1 was found in both SCGs and MCGs. We apologize for that misleading data representation and we have changed our figures accordingly (Figure 4 and Supplementary Fig. 5). Furthermore, we clarified that point in our result section page 7 lines 171 – 175 ("Both granule types ... (Supplementary Fig. 6)").

In addition, we tried to find a good candidate for degranulation marker for MCGs that is present on the organelle membrane. One candidate could be the calcium-independent mannose-6-phosphate receptor (CI-M6PR; IGF2R; position 112 on the ranking curve) present only on MCGs. However, to the best of our knowledge there is no good anti-IGF2R antibody specific for mouse on the market, which is why we cannot test it.

Comment 3: "LysoTracker is also often used for live cell studies of lytic granules. It would very valuable to know if lysotracker labels both type of granules or only SCGs?"

Reply:

We performed this experiment by loading LysoTracker-red in Gzmb KI T cells, which we also labelled with WGA-647 to mark MCGs. In Supplementary figure 6 we now show that LysoTracker labelled both MCGs and SCGs in CTLs.

Comment 4: "Lytic granules are known to have very low pH. Did you compare pH in both types of SCG and MCG?"

Reply:

From the experiment with LysoTracker (see comment 3), which accumulates specifically in all acidic compartments, we can conclude that both MCG and SCG are acidic granules. We implemented this finding in the result section page 7 lines 171 – 175 ("Both granule types ... (Supplementary Fig. 6)"). However, their exact pH value is not resolved from this experiment. We have measured the overall pH of the cytotoxic granules population in mouse CD8⁺ cells before (Chitirala et al, 2020). Nevertheless, in these experiments it was not possible to measure the pH of individual granules. This would be a prerequisite to be able to measure separately the pH of MCG and SCG. The technical difficulties to resolve individual granule's pH in living cells are measurement's sensitivity and resolution, and acquisition speed as granules are highly mobile. While we agree that this is a very interesting question, we are currently not able to fully address this point in a timely manner and believe that it is out of the scope of our current manuscript. We will address this very interesting question in the future.

Comment 5: "From the presented data, it seems that the majority of CGs in the cell are surprisingly MCGs and MCGs also represent the majority of secreted granules. What is the amount of Perforin and Granzyme B per granule volume in SCGs compared to MCGs?"

Reply:

To address this comment we designed an experiment in which we observed individual isolated SCGs and MCGs with using super resolution stimulated emission depletion (STED)

microscopy. For that we used sucrose gradients fraction 6 and 8 isolated from Synaptobrevin2-mRFP KI mouse CTLs, which were carefully settled on gelatine coated coverslips after mild PFA fixation. This ensured that the granules remained intact. We further labelled mRFP on granule membrane and GzmB with primary antibodies and STED compatible secondary antibodies to analyse GzmB content and its distribution within granules. We observed a strong punctate GzmB staining in MCGs while in SCG the staining was weaker and more diffuse (new Figure 5g). The analysis of individual granules revealed that SMAPs contain a very high load of GzmB. While the volume occupied by GzmB in both type of granule was similar, the GzmB concentration in MCGs is 20 % higher than in SCGs (new Fig. 5h). We added the following paragraph in the result section page 9 – 10 line 226 – 238 (“To evaluate the potential killing ... target cells.”). Regarding Perforin, we have tested many different antibodies but none of them appeared to be specific in mouse CTL immunofluorescence. Therefore, we could not repeat this experiment with the detection of Perforin. We completed the material and method section on page 20 – 21, line 522 – 550 (“STED microscopy ... presentation purpose.”) accordingly.

Comment 6: “Could you comment on the contribution of both types of CGs to the killing capacity of the cell?”

Reply:

To address the functionality of MCGs and SCGs to kill target cells, we conducted a killing assay to evaluate their cytotoxicity. We settled p815 target cells down on coverslips previously coated with MCGs or SCGs after high speed centrifugation of sucrose gradient fractions 6 and 8, respectively. We assessed target cell apoptosis through the cleavage of caspase3 in a time dependent manner (new Fig. 6e). We observed under our experimental conditions, that both, SCGs and MCGs, can induce target cell apoptosis within 6 h (new Fig. 6f). In long term (19h), MCGs were more efficient to kill target cells as SCGs, most likely due to the concentrated amount of cytotoxic proteins stored in SMAPs. We added these data to the result section page 10 – 11, line 255 – 265 (“We further assessed ... are released upon IS formation.”), we completed the material and method section on page 19, line 490 – 499 (“Evaluation of MCG and SCG cytotoxicity ... evaluate cell death.”) accordingly, and added in the discussion page 12, line 298 - 300 the following sentence: “MCGs as well as SCGs can kill target cells (Fig. 6e), and MCGs appear to contain more GzmB than SCGs (Fig. 5h).”.

Reviewer #2

The study by Chang and colleagues identifies a novel cytotoxic granule component of CD8+ T cells that appears to be a major delivery mechanism of cytotoxic molecules. This is an impressive study that not only reconciles earlier descriptions of cytotoxic vesicles (SMAPs), but demonstrates that these granules (MCG vs SCGs) not only have distinct components but also likely form from distinct biogenesis. The observations in these studies will be well received by the field and open up new avenues of research by providing not only insights into the fact there are distinct classes of cytotoxic granules, but the different composition of these granules.

The manuscript is well written, with the conclusions well supported by the data and their interpretation. To that end, there are only a few issues that perhaps would help clarify some questions that came to mind during the review.

Comment 1: “While the focus is on the role of cytolytic activity, there has been prior reports suggesting that cytolytic granules, particularly Granzyme B, have a role in CD8⁺ T cell trafficking through tissue (Prakesh et al., *Immunity*. 2014 Dec 18;41(6):960-72). Is there any indication that perhaps rather than both MCGs and SCGs having a role in cytolytic activity, there might be a role for either (or both) in transmigration through tissue based on protein composition?”

Reply:

We appreciate the reviewer comment and we thank for pointing out this highly interesting aspect of general T cell function in the immunological response. We have not yet conducted any experiments related to cell adhesion and migration to look at the contribution from MCG and SCG granule population. From our proteomics data, we found that the collection of integrins that are present in both SCGs and MCGs (modified Fig. 4b) have not yet been described to play a role in cell migration. Chemokines, which are known guidance molecules, were not found on these granules except for CCR7 on SCG (at non-significant level). However, CCR7 seems not to contribute directly to transmigration but rather T cell homing. We currently have no indication if either type of granule plays a role in transmigration.

Comment 2: “While reassuring to see GrzB and Pfp present in these granules, there was an absence of other granzyme family members (eg GrzA, K, M). Is there a reason for this (type of CTL used)?”

Reply:

The reviewer correctly pointed out that there are other granzymes in mouse. It has been published that murine CTLs contain 10 different granzymes (A-G, K, M and N) (Ewen et al., 2012 (doi.org/10.1038/cdd.2011.153)), of which GzmA and GzmB are the main cytotoxic proteins (Kaiserman et al., 2006 (doi.org/10.1083/jcb.200606073)). In our study, we used CD3/CD28 bead activated mouse CD8⁺ T-cells that had been in culture for 3-4 days for the organelle isolation. Under these conditions, we found by mass spectrometry that GzmB is the main granzyme expressed in SCGs and MCGs. Additionally, we also found that six peptides of GzmA could be detected in SCG population with no significance. Our finding is in agreement with several studies that also showed that GzmB is the most prominent granzyme expressed in mouse CTLs at day 3 to 4 in vitro (Kelso et al. (2002, doi: 10.1093/intimm/14(12):1288-1294); Cai et al. (2009, doi: 10.4049/jimmunol.0804333)). None the less, to strengthen our finding that the most abundant GzmB is fully functional and can induce target cell death we performed a cytotoxicity assay by using P815 cell line. We found that isolated SCGs and MCGs induced target cell apoptosis through the caspase-3 death pathway (Fig. 6f). Since this pathway is induced primarily by GzmB, our result show that it is not only the most abundant but also a very effective cytotoxic component of our granules.

Comment 3: “A quick question, Fraction 6 (MCG) is distinguished from Fraction 8 by Na⁺/K⁺ ATPase in figure 1. It was identified in the mass spec data but not highlighted in panel b, figure 4. Adding this might help with orientating readers.”

Reply:

In the quantitative western blot characterizing sucrose fractions 2 to 10 in Fig. 1d the Na⁺/K⁺-ATPase was only used to identify the presence of plasma membrane proteins as a contamination. After immune-isolation to enrich Syb2 containing cytotoxic granules, the representative western blot for IP6 and IP8 showed the presence of Na⁺/K⁺-ATPase in the

input and supernatant, but strongly reduced in the immune-precipitation of both fractions. Na⁺/K⁺ ATPase subunits such as alpha-1 and beta 3 are present in both, MCGs and SCGs. We have made this point clearer in the text results page 5, line 114 – 116 (“On the other hand...in both supernatants (Fig. 2b).”).

Comment 4: “TCR CD3 subunits, among other membrane proteins, are contained within MCGs. Is this due to the enveloped nature of some of the cores within organelle, or is there perhaps some regulatory function of these MCGs in regulating T cell activation?”

Reply:

At this point we can only envision that recycling TCR CD3 subunits (d, e, g, zeta) could integrate the cytotoxic granule biogenesis pathway and appear as a component of SCG and MCG (Supplementary Table 1; mass spectrometry data). Interestingly CD3 subunits (e, g and zeta) were more abundant in MCGs than in SCGs. We speculate that CD3 subunits are sorted in a differential manner to the two granule types and have different regulatory functions. For example, SCGs could play an important role in delivering TCR CD3 subunits to the plasma membrane. Whereas CD3 contained in MCGs is most probably localized to T cell-derived exosomes (Blachard et al., 2002 (doi: 10.4049/jimmunol.168.7.3235), which we also identified in MCGs (see discussion page 12, Line 294 – 296). Accordingly, the TCR CD3 subunits on the exosomes might be released during MCG exocytosis and they could play a regulatory role in an auto or paracrine manner. However, the function of T-cell derived exosomes is still enigmatic.

Comment 5: “While SMAPs have been described for NK cells, what isn't clear is whether NK cells have the same heterogeneity in cytolytic granules (MCGs vs SCGs)? Some discussion of this would be worthwhile.”

Reply:

To address this question, we have looked at cellular organelles in human NK92 cell line and have indeed found heterogeneity of granules in these cells. By using cryo-soft-X-ray tomography we found that NK92 cells contain two classes of organelles that share the same morphological features with those of mouse T cells and had comparable sizes for SCGs and MCGs. We believe these are the same two granule populations. We integrated these data in the result part (Supplementary Fig. 4), and addressed this point in the result page 6-7 Line 153 – 159 (“Detailed analysis ... Supplementary Video 1 and 2.”) and in the discussion page 11, line 273 - 278 (“Interestingly, we found ... mediated killing.”).

Reviewer #3

This paper describes the observation of two classes of fusion-competent granules, single core granules (SCGs) and multi core granules (MCGs), and defining their size, morphology and protein composition. In addition, functional analyses of these fusion-competent granules were conducted and it was shown that both classes of granules fuse with the plasma membrane at the IS.

Thus a novel class of CGs named MCGs was discovered based on morphological appearance. MCGs could be a future target for modulating T cell killing efficiency in immunotherapy however this study was not conducted yet.

This paper, though describes important findings on mechanism by which CTLs kill their targets, would be a better fit for immunology related journals.

Reply:

We appreciate the reviewer comment on MCG killing ability. It is indeed an essential point of the manuscript. We have assessed target cell apoptosis through the cleavage of caspase3 in a time dependent manner and observed under our experimental conditions, that both, SCGs and MCGs, can induce apoptosis within 6 h (new Fig. 6f and 6e). In long term (19h), MCGs were more efficient to kill target cells as SCGs, most likely due to the concentrated amount of cytotoxic proteins stored in SMAPs. We added these data to the result section page 10 – 11, line 255 – 265 (“We further assessed ... are released upon IS formation.”), we completed the material and method section on page 19, line 490 – 499 (“Evaluation of MCG and SCG cytotoxicity ... evaluate cell death.”) accordingly, and added in the discussion page 12, line 298 - 300 the following sentence: “MCGs as well as SCGs can kill target cells (Fig. 6e), and MCGs appear to contain more GzmB than SCGs (Fig. 5h).”.

Furthermore, we show an entirely new set of data, in which we found that human NK cells also contain two similar types of cytolytic granules like MCGs and SCGs by Cryo-Soft-X-ray tomography analysis. We integrated these data in the manuscript, page 6-7 Line 153 – 159 and as a new supplementary figure 4 and Supplementary Video 1 and 2. We expanded our material and method section accordingly (see page 14 and, Line 357 – 361 and page 24, Line 626 – 633). We also commented on these findings in the discussion page 11, line 273 - 278 (“Interestingly, we found ... mediated killing.”). This finding enhances the scientific value of our manuscript as human NK cells are the potential cell type for immunotherapy in cancer treatment. Thus our results are highly relevant for a very broad audience.

Additionally, our work will raise attention for cell biologists as it can be potentially transferred to neuronal cells due to the similarity of immune synapse and neuronal synapse as mentioned in the discussion (page 13, line 320 - 331). In neuronal cells, the small clear vesicles (SCVs) and dense core vesicles (DCVs) contain neurotransmitters or neuromodulators, respectively that act in a different time scale and target.

Hence, our manuscript would attract the attention of a wide range of readers such as oncologists, neurologists and cell biologists, which is why we are confident that nature communication is a perfect fit for presenting our data.

REVIEWERS' COMMENTS

Reviewer #1 (Remarks to the Author):

All points have been addressed. I thank the authors for the extra effort!
I have no further comments.

Reviewer #2 (Remarks to the Author):

The authors should be commended on addressing the majority of my concerns. I have only one minor point. The authors may not have quite understood my point regarding the study by Prakesh et al., (*Immunity*, 41:960, 2014). In this study, the authors showed that rather than cytolytic killing, GrzB was in fact important for transmigration of CTL through parenchymal tissue. The substrates for This biological role is cleavage of basement membrane components enabling chemokine mediated transmigration. This paper should be cited in the discussion given the description of distinct vesicles identified by the authors in this study.

We appreciate the efforts that the reviewers made on their careful reviews, enthusiasm for our work. We further thank the editors for the positive feedback to revise the manuscript. The changes made in response to reviewer2's concern is listed below and highlighted (green) in the manuscript.

Reviewer #1 (Remarks to the Author):

All points have been addressed. I thank the authors for the extra effort!
I have no further comments.

Reply:

Thanks again for the constructive suggestions and reviewing our work.

Reviewer #2 (Remarks to the Author):

The authors should be commended on addressing the majority of my concerns. I have only one minor point. The authors may not have quite understood my point regarding the study by Prakesh et al., (Immunity, 41:960, 2014). In this study, the authors showed that rather than cytolytic killing, GrzB was in fact important for transmigration of CTL through parenchymal tissue. The substrates for This biological role is cleavage of basement membrane components enabling chemokine mediated transmigration. This paper should be cited in the discussion given the description of distinct vesicles identified by the authors in this study.

Reply:

Reviewer had a very good point of diverse function of GzmB in T cells. We added two sentences accordingly in the discussion section in page 12, line 302-304 ("In addition to cytolytic killing,... from MCGs.") and cited this reference.